# RNA-FrameFlow:
# Flow Matching for de novo 3D RNA Backbone Design

**Rishabh Anand**[*,1]    **Chaitanya K. Joshi**[*,2]    **Alex Morehead**[3]    **Arian R. Jamasb**[4]
**Charles Harris**[2]    **Simon V. Mathis**[2]    **Kieran Didi**[5]    **Rex Ying**[1]    **Bryan Hooi**[6]    **Pietro Liò**[2]

[*]*Equal contribution*    [1]*Yale University*    [2]*University of Cambridge (UK)*    [3]*Lawrence Berkeley National Laboratory*
[4]*Prescient Design, Genentech, Roche*    [5]*University of Oxford (UK)*    [6]*National University of Singapore*

*Correspondence to:* `rishabh.anand@yale.edu, chaitanya.joshi@cl.cam.ac.uk`

**Reviewed on OpenReview:** *https://openreview.net/forum?id=wOc1Yx5s09*

## Abstract

We introduce RNA-FRAMEFLOW, the first generative model for 3D RNA backbone design. We build upon $SE(3)$ flow matching for protein backbone generation and establish protocols for data preparation and evaluation to address unique challenges posed by RNA modeling. We formulate RNA structures as a set of rigid-body frames and associated loss functions which account for larger, more conformationally flexible RNA backbones (13 atoms per nucleotide) vs. proteins (4 atoms per residue). Toward tackling the lack of diversity in 3D RNA datasets, we explore training with structural clustering and cropping augmentations. Additionally, we define a suite of evaluation metrics to measure whether the generated RNA structures are globally self-consistent (via inverse folding followed by forward folding) and locally recover RNA-specific structural descriptors. The most performant version of RNA-FRAMEFLOW generates locally realistic RNA backbones of 40-150 nucleotides, over 40% of which pass our validity criteria as measured by a self-consistency TM-score $\geq 0.45$, at which two RNAs have the same global fold. Open-source code: `github.com/rish-16/rna-backbone-design`

## 1 Introduction

**Designing RNA structures.** Proteins, and the diverse structures they can adopt, drive essential biological functions in cells. Deep learning has led to breakthroughs in structural modeling and design of proteins (Jumper et al., 2021; Dauparas et al., 2022; Watson et al., 2023), driven by the abundance of 3D data from the Protein Data Bank (PDB). Concurrently, there has been a surge of interest in *Ribonucleic Acids* (RNA) and RNA-based therapeutics for gene editing, gene silencing, and vaccines (Doudna & Charpentier, 2014; Metkar et al., 2024). RNAs play several roles in cells: they carry genetic information coding for proteins (mRNA) as well as perform functions driven by their tertiary structural interactions (riboswitches and ribozymes). While there is growing interest in designing such structured RNAs for a range of applications in biotechnology and medicine (Mulhbacher et al., 2010; Damase et al., 2021), the current toolkit for 3D RNA design uses classical algorithms and heuristics to assemble RNA motifs as building blocks (Han et al., 2017; Yesselman et al., 2019). However, hand-crafted heuristics are not always broadly applicable across multiple tasks and rigid motifs may not fully capture the conformational dynamics that govern RNA functionality (Ganser et al., 2019; Li et al., 2023b). This presents an opportunity for deep generative models to learn data-driven design principles from existing 3D RNA structures.

**What makes deep learning for RNA hard?** The primary challenge is the paucity of raw 3D RNA structural data, manifesting as an absence of ML-ready datasets for model development (Joshi et al., 2025). Protein structure is primarily driven by hydrogen bonding along the backbone, and current deep learning models incorporate this inductive bias through backbone frames to represent residues (Jumper et al., 2021).

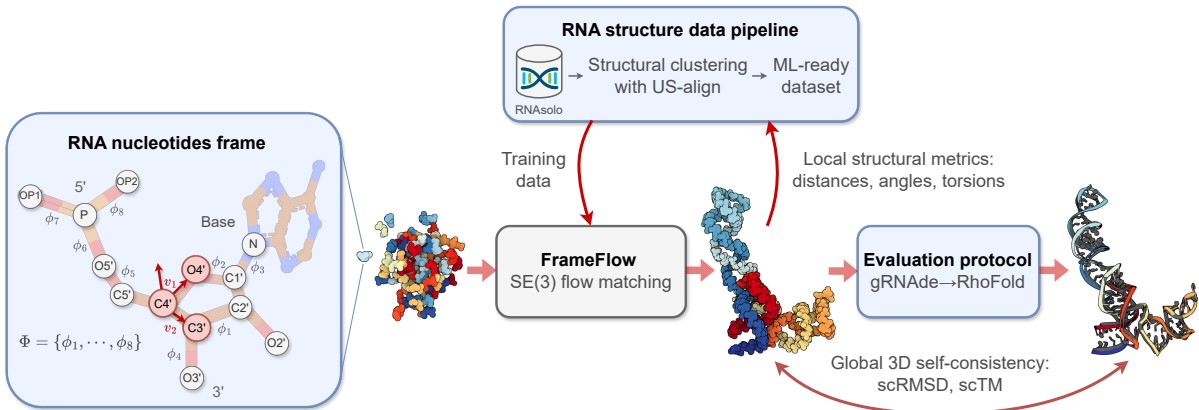

Figure 1: **The RNA-FrameFlow pipeline for 3D backbone generation.** Our implementation establishes RNA-specific protocols for data preparation and evaluation for FrameFlow (Yim et al., 2023a). (1) Each nucleotide in the RNA backbone is converted into a *frame* to parameterize the placement of $C4'$ by a translation, $C3' - C4' - O4'$ by a rotation, and the rest of the atoms via 8 torsion angles $\Phi$. (2) We train generative models on all RNA structures of length 40-150 nucleotides from RNAsolo (Adamczyk et al., 2022). We also explore training with structural clustering and cropping augmentations to tackle the lack of diversity in 3D RNA datasets. (3) We introduce evaluation metrics to measure the recovery of local structural descriptors and global self-consistency of designed structures via inverse-folding with gRNAde (Joshi et al., 2025) followed by forward-folding with RhoFold (Shen et al., 2022).

RNA structure, however, is often more conformationally flexible and driven by base pairing interactions across strands as well as base stacking between rings of adjacent nucleotides (Vicens & Kieft, 2022).

Additionally, RNA nucleotides, the equivalent of amino acids in proteins, include significantly more atoms as part of the backbone (13 compared to 4) which necessitates a generalization of backbone frames where the placement of most atoms needs to be parameterized by torsion angles. These complexities have contributed to relatively poor performance of deep learning for RNA structure prediction compared to proteins (Kretsch et al., 2023; Abramson et al., 2024). Additionally, structure prediction models cannot directly be used for designing or generating *novel* RNA structures with desired constraints. Thus, the lack of *de novo* RNA design models presents a gap worth addressing, which we aim to do in this work.

**Our Contributions.** We develop RNA-FRAMEFLOW, the first generative model for 3D RNA backbone design, which trains on RNA of length 40-150 from the PDB; see Figure 1. We devise RNA frames to capture the significantly larger RNA nucleotides while still supporting the modelling of all backbone atoms implicitly. Our method can unconditionally sample plausible backbones with over 40% validity, as measured by our custom evaluation pipeline that computes global self-consistency and local structural measurements. We also explore data augmentation protocols to address the paucity of diverse RNA structural data, which improves the novelty of designed RNA. We hope this work stimulates future research in generative models for RNA design.

## 2  The RNA-FrameFlow Pipeline

We are concerned with building a generative model that unconditionally outputs 3D RNA backbone structures, sampled from a distribution of realistic structures. Formally, given an RNA sequence of $N_{nt}$ nucleotides, we aim to generate a real-valued tensor $\mathbf{X}$ of shape $N_{nt} \times 13 \times 3$ representing 3D coordinates for 13 backbone atoms per nucleotide. In the following sections, we will describe how we adapt FrameFlow (Yim et al., 2023a), an $SE(3)$ equivariant flow matching model for protein backbones, to our setting.

### 2.1 Representing RNA Backbones as Frames

**Rationale for RNA frames.** In proteins, it is standard to represent each residue by a frame centered at $C_\alpha$ with vectors along $C_\alpha - N$ and $C_\alpha - C$, and $O$ is placed assuming an idealized planar geometry (Jumper et al., 2021). However, no such canonical frame representation exists for RNAs. Furthermore, the RNA nucleotide contains significantly more backbone atoms (13) compared to proteins (4). As shown in Figure 1, the backbone fragment of RNA nucleotides comprise a phosphate group ($P, OP1, OP2, O5'$), a ribose sugar ($C1' - C5', O2', O3', O4'$), and a nitrogen atom $N$ at the stem of the base. To avoid the high modelling complexity of explicitly generating coordinates for all 13 backbone atoms, we decide to represent the atoms within each nucleotide as a rigid-body frame; this enables inferring the positions of all intra-nucleotide atoms via a frame center and orientation.

**Constructing RNA frames.** We select the $C4', C3'$, and $O4'$ atoms to create the frame for each nucleotide, as in Morehead et al. (2023). All other backbone atoms are inferred with 8 torsions $\Phi = \{\phi_i\}_{i=1}^8$, $\phi_i \in SO(2)$ that are predicted post-hoc after frame generation. The Gram-Schmidt process is used on $v_1, v_2$ defined by the vectors along the $C4' - O4'$ and $C4' - C3'$ bonds; $C5'$ is imputed based the positions of the other 3 atoms and tetrahedral geometry. Given the 8 torsion angles, we autoregressively place non-frame atoms in order of the torsions $\Phi$ in Figure 1, constructing the final set of *all-atom* RNA nucleotides. We describe this imputation of non-frame atoms as well as the choice of torsion angle parameterization in Appendix A.4.

**Choice of frame atoms.** We consider two main factors for selecting the atoms to create RNA frames: (1) atoms should have roughly the same spatial orientation w.r.t. each other; and (2) atoms should be reasonably close to the centroid in the nucleotide to reduce error accumulation when placing the furthest non-frame atoms. We choose $\{C3', C4', O4'\}$ as these atoms have relatively lower atomic displacement in natural RNA (Harvey & Prabhakaran, 1986); see Appendix C.5 for empirical evidence on atomic displacements. The remaining non-frame backbone atoms in the ribose ring and the phosphate group are parameterized by torsion angles to account for their relative conformational flexibility. We also believe this choice of frame enables models to capture *ring puckering*, by which the five ribose atoms contort the ideal plane of the sugar ring to alleviate steric strain. This also influences how RNA interacts with partners to form complexes (Clay et al., 2017); see Appendix C.6 for additional details and results on puckering.

### 2.2 $SE(3)$ Flow Matching on RNA Frames

**Input.** Given a set of 3D coordinates, a simultaneous rotation and translation $(r, x)$ forms an orientation-preserving rigid-body transformation of the coordinates. The set of all such transformations in 3D is the Special Euclidean group $SE(3)$, which composes the group of 3D rotations $SO(3)$ and 3D translations in $\mathbb{R}^3$. We can represent an RNA frame $T = (r, x)$ as a translation $x \in \mathbb{R}^3$ from the global origin to place $C4'$ and a rotation $r \in SO(3)$ to orient $C3' - C4' - O4'$. Compared to working with raw 3D coordinates for each backbone atom, using the frame representation entails performing flow matching on the space of $SE(3)$. This is an inductive bias to reduce the degrees of freedom the generative model needs to learn. Instead of predicting 13 correlated 3D coordinates independently (39 quantities) for each nucleotide, we instead predict one 3D coordinate (of $C4'$) and one 3×3 rotation matrix (12 quantities). We follow Chen & Lipman (2024) and Yim et al. (2023a)'s framework for flow matching on $SE(3)$, which we summarise subsequently.

**Overview.** Flow matching generates or learns how to place and orient a set of $N$ frames $\mathbf{T} = \{T^{(n)}\}_{n=1}^N$, where $T^{(n)} = (r^{(n)}, x^{(n)})$, to form an RNA backbone of length $N$. To do so, we initialize frames at random in 3D space at time $t = 0$, and train a denoiser or flow model to iteratively refine the location and orientation of each frame for a specified number of steps until time $t = 1$.

Suppose $p_0(T_0)$ and $p_1(T_1)$ are the marginal distributions of randomly oriented and ground truth frames from our dataset of RNA structures, respectively. Suppose a time-dependent vector field $u_t$ leads to an ODE between the two distributions $p_0$ and $p_1$, i.e., assume there is a way to map from noisy samples to the corresponding true samples. This solution forms a ground truth *probability path* $p_t$ between the two distributions at time $t \in [0, 1]$, which we can use to transform noisy samples to the true distribution. The *continuity equation* $\frac{\partial p}{\partial t} = -\nabla \cdot (p_t u_t)$ relates the vector field $u_t$ to the evolution of the probability path $p_t$.

Given a noisy frame $T_0$ sampled from $p_0(T_0)$ and the corresponding ground truth frame $T_1$ sampled from $p_1(T_1)$, we construct a *flow* $T_t$ by following the probability path $p_t$ between $T_0$ and $T_1$ for any time step $t$ sampled from $\mathcal{U}(0, 1)$. As shown by Chen & Lipman (2024) for the $SE(3)$ group (and other manifolds), the geodesic between the states $T_0$ and $T_1$ can be used to define an interpolation:

$$T_t = \exp_{T_0}(t \cdot \log_{T_0}(T_1)). \tag{1}$$

Here, $\exp(\cdot)$ and $\log(\cdot)$ are the *exponential* and *logarithmic* maps that enable moving (taking random walks) on curved manifolds such as the $SE(3)$ group. As we can decompose a frame $T = (r, x)$ into separate rotation and translation terms, we can obtain closed-form interpolations for the group of rotations $SO(3)$ and translations $\mathbb{R}^3$. This gives us two independent flows:

$$\text{Translations:} \quad x_t = tx_1 + (1 - t)x_0 \ , \qquad \text{Rotations:} \quad r_t = \exp_{r_0}(t \cdot \log_{r_0}(r_1)) \ . \tag{2}$$

The random translation $x_0$ is sampled from a zero-centered Gaussian distribution $\mathcal{N}(0, \mathbf{I})$ in $\mathbb{R}^3$, and the random rotation $r_0$ is sampled from $\mathcal{U}(SO(3))$, a generalization of the uniform distribution for the group of rotations, $SO(3)$. For an RNA backbone consisting of a set of $N$ frames $\mathbf{T} = \{ T^{(n)} \}_{n=1}^N$, we can define the interpolation for each frame in parallel via the aforementioned procedure.

**Training.** During training, we would like to learn a parameterized vector field $v_\theta(\mathbf{T}_t, t)$, a deep neural network with parameters $\theta$, which takes as input the intermediate frames $\mathbf{T}_t$ at time $t$ sampled from $\mathcal{U}(0, 1)$, and predicts the final frames $\hat{\mathbf{T}} = \{\hat{T}^{(n)}\}_{n=1}^N$, where $\hat{T}^{(n)} = (\hat{r}_t^{(n)}, \hat{x}_t^{(n)})$. The ground truth vector field $u_t$ for mapping from the intermediate frames $\mathbf{T}_t$ to the ground truth frames $\mathbf{T}_1$ can also be decomposed into a ground truth rotation and translation for each frame $T^{(n)}$:

$$\text{Translations:} \quad u_t(x^{(n)}|x_0^{(n)}, x_1^{(n)}) = x_1^{(n)} \ , \qquad \text{Rotations:} \quad u_t(r^{(n)}|r_0^{(n)}, r_1^{(n)}) = \log_{r_t^{(n)}}(r_1^{(n)}) \ . \tag{3}$$

To train the model $v_\theta$, we compute separate losses for the predicted rotation $\hat{r}_t \in SO(3)$ and translation $\hat{x}_t \in \mathbb{R}^3$. The combined $SE(3)$ flow matching loss over $N$ frames is as follows:

$$\mathcal{L}_{SE(3)} = \mathbb{E}_{t, \ p_0(\mathbf{T}_0), \ p_1(\mathbf{T}_1)} \left[ \frac{1}{(1-t)^2} \sum_{n=1}^N \underbrace{\left\| \hat{x}_t^{(n)} - x_1^{(n)} \right\|_{\mathbb{R}^3}^2}_{\mathcal{L}_{\mathbb{R}^3}^{(n)}} + \underbrace{\left\| \log_{r_t^{(n)}}(\hat{r}_1^{(n)}) - \log_{r_t^{(n)}}(r_1^{(n)}) \right\|_{SO(3)}^2}_{\mathcal{L}_{SO(3)}^{(n)}} \right]. \tag{4}$$

The architecture of the flow model $v_\theta$ is similar to the structure module from AlphaFold2 comprising Invariant Point Attention layers interleaved with standard Transformer encoder layers, following Yim et al. (2023a;b). We use an MLP head to predict torsion angles $\Phi$.

**Auxiliary losses.** The inclusion of auxiliary loss terms to the objective in Equation 4 can be seen as a form of adding domain knowledge into the training process (Yim et al., 2023b). We include 3 additional losses that operate on all the backbone atoms from the predicted frames, weighted by tunable coefficients to modulate their contribution to the total loss:

$$\mathcal{L}_{\text{tot}} = \mathcal{L}_{SE(3)} + \mathcal{L}_{\text{bb}} + \mathcal{L}_{\text{dist}} + \mathcal{L}_{\text{tors}} \ . \tag{5}$$

Suppose $S = \{C4', C3', O4'\}$ is the set of frame atoms[1] and the sequence length is $N$. We summarise the auxiliary losses subsequently.

- **Coordinate MSE $\mathcal{L}_{\text{bb}}$:** A direct MSE is computed between generated and ground truth coordinates. Here, $a, \hat{a}$ are the ground truth and predicted atomic coordinates for the frame atoms:

$$\mathcal{L}_{\text{bb}} = \frac{1}{|S|N} \sum_{n=1}^N \sum_{a \in S} \|a^{(n)} - \hat{a}^{(n)}\|^2. \tag{6}$$

---

[1]In Appendix B.1, we show how including more backbone atoms better accounts for larger RNA nucleotides and improves validity of generated samples.

- **Distogram loss $\mathcal{L}_{\text{dist}}$**: A distogram $D \in \mathbb{R}^{NS \times NS}$ containing all-to-all coordinate differences between the atoms in an RNA structure is computed. Let $D_{ab}^{(nm)} = \|a^{(n)} - b^{(m)}\|$ be the elements of the distogram for the ground truth structure. Here, atom $a$ belongs to nucleotide $n$ and atom $b$ to nucleotide $m$. Given the corresponding predicted distogram $\hat{D}_{ab}^{(nm)}$, we compute another difference between the tensors:

$$\mathcal{L}_{\text{dist}} = \frac{1}{(|S|N)^2 - N} \sum_{\substack{n,m=1 \\ n \neq m}}^{N} \sum_{a,b \in S} \|D_{ab}^{(nm)} - \hat{D}_{ab}^{(nm)}\|^2. \tag{7}$$

- **Torsional loss $\mathcal{L}_{\text{tors}}$**: An angular loss between the 8 predicted torsions by the auxiliary MLP head and the angles from the ground truth structure with all backbone atoms. Suppose $\phi \in \Phi_n$ and $\hat{\phi} \in \hat{\Phi}_n$ are the ground truth and predicted torsion angles for residue $n$, we compute:

$$\mathcal{L}_{\text{tors}} = \frac{1}{8N} \sum_{n=1}^{N} \sum_{\phi \in \Phi_n} \left( \|\phi - \hat{\phi}\|^2 \right). \tag{8}$$

**Sampling.** To generate or unconditionally sample an RNA backbone of length $N$, we initialize a random point cloud of frames. We use our trained flow model $v_\theta$ within an ODE solver to iteratively transform the noisy frames into a realistic RNA backbone. For each nucleotide, we begin with a noisy frame $T_0 = (r_0, x_0)$ at time step $t = 0$, and integrate to $t = 1$ using the Euler method for a specified number of steps $N_T$, with step size $\Delta t = 1/N_T$. At each step $t$, the flow model $v_\theta$ predicts updates for the frames via a rotation $\hat{r}_1$ and translation $\hat{x}_1$:

$$\text{Translations:} \quad x_{t+\Delta t} = x_t + \Delta t \cdot (\hat{x}_1 - x_t), \tag{9}$$

$$\text{Rotations:} \quad r_{t+\Delta t} = \exp_{r_t}(c \, \Delta t \cdot \log_{r_t}(\hat{r}_1)), \tag{10}$$

where $c =$ is a tunable hyperparameter governing the exponential sampling schedule for rotations.

**Conditional generation.** The unconditional sampling strategy described above aims to generate realistic RNA backbones sampled from the training distribution. However using generative models in real-world design tasks entails *conditional* generation based on specified design constraints or requirements (Ingraham et al., 2022; Watson et al., 2023), which we are currently exploring. For example, unconditional models can leverage inference-time guidance strategies (Wu et al., 2024), be fine-tuned conditionally (Denker et al., 2024) or in an amortized fashion for motif-scaffolding (Didi et al., 2023). For sequence conditioning and structure prediction, embeddings from language models can also be incorporated (Penic et al., 2024; He et al., 2024).

## 3 Experiments

**3D RNA structure dataset.** RNAsolo (Adamczyk et al., 2022) is a recent dataset of RNA 3D structures extracted from isolated RNAs, protein-RNA complexes, and DNA-RNA hybrids from the Protein Data Bank (as of January 5, 2024). The dataset contains 14,366 structures at resolution $\leq 4$ Å (1 Å = 0.1nm). We select sequences of lengths between 40 and 150 nucleotides (5,319 in total) as we envisioned this size range contains structured RNAs of interest for design tasks.

**Evaluation metrics.** We evaluate our models for unconditional RNA backbone generation, analogous to recent work in protein design (Yim et al., 2023b;a; Bose et al., 2023; Lin & AlQuraishi, 2023); see Figure 2. We generate 50 backbones for target lengths sampled between 40 and 150 at intervals of 10. We then compute the following indicators of quality for these backbones:

- **Validity (scTM $\geq$ 0.45)**: We inverse fold each generated backbone using gRNAde (Joshi et al., 2025) and pass $N_{\text{seq}} = 8$ generated sequences into RhoFold (Shen et al., 2022). We then compute the self-consistency TM-score (scTM) between the predicted RhoFold structure and our backbone at the $C4'$ level. We say a backbone is *valid* if scTM $\geq$ 0.45; this threshold corresponds to roughly the same fold between two RNAs (Zhang et al., 2022). Alternatively, we use an RMSD threshold of 4.3 Å, corresponding to the median RMSD of RhoFold on RNAsolo sequences.

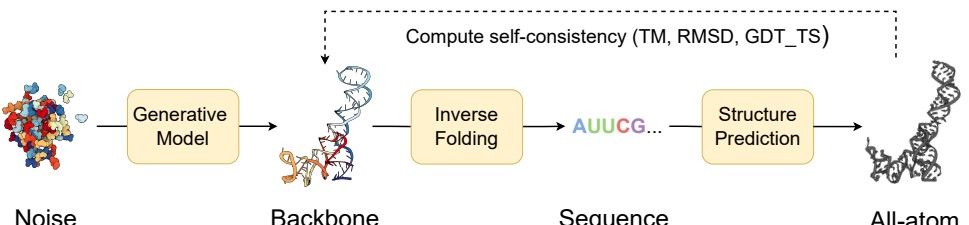

Figure 2: **Structural self-consistency evaluation.** We sample a backbone from our model and pass it through an inverse folding model (gRNAde) to obtain $N_{\text{seq}} = 8$ sequences. Each sequence is fed into a structure prediction model (RhoFold) to get the predicted all-atom backbone. Self-consistency between each predicted backbone and the generated sample is measured with TM-score (we also report RMSD and GDT_TS). For a given generated sample, we thus have $N_{\text{seq}} = 8$ TM-scores of which we take the maximum as the `scTM` score for that sample.

- **Diversity**: Among the valid samples, we compute the number of unique structural clusters formed using `qTMclust` (Zhang et al., 2022) and take the ratio to the total number of samples. Two structures are considered similar if their TM-score $\geq 0.45$. This metric shows how much each generated sample varies from others across various sequence lengths.

- **Novelty**: Among the valid samples, we use `US-align` (Zhang et al., 2022) at the $C4'$ level to compute how structurally dissimilar the generated backbones are from the training distribution. For a set of samples for a given sequence length, we compute the TM-score between all pairs of generated backbones and training samples, and for each generated backbone, we assign the highest TM-score. We call the average across this set, `pdbTM`.

- **Local structural measurements**: We measure the similarity between bond distances, bond angles, and dihedral angles from the set of generated samples and the training set. To do so, we compute histograms for each of the local structural metrics and use 1D Earth Mover's distance to measure the similarity between generated and training distributions.

**Hyperparameters.** We use 6 IPA blocks in our flow model, with an additional 3-layer torsion predictor MLP that takes in node embeddings from the IPA module. Our final model contains 16.8M trainable parameters. We use AdamW optimizer with learning rate 0.0001, $\beta_1 = 0.9$, $\beta_2 = 0.999$. We train for $120K$ gradient update steps on four NVIDIA GeForce RTX 3090 GPUs for $\sim$18 hours with a batch size $B = 28$. Each batch contains samples of the same sequence length to avoid padding. Further hyperparameters are listed in Appendix A.1.

## 4 Backbone Generation Results

### 4.1 Global evaluation of generated RNA backbones

We begin by analyzing RNA-FRAMEFLOW's samples using the aforementioned evaluation metrics. For validity, we report percentage of samples with `scTM` $\geq 0.45$; for diversity, we report the ratio of unique structural clusters to total **valid** samples; and for novelty, we report the highest average `pdbTM` to a match from the PDB. For each sequence length between 40 and 150, at intervals of 10, we generate 50 backbones. Table 1 reports these metrics across different variants for the number of denoising steps $N_T$. The average `scTM` and `scRMSD` of *valid* samples are $0.641 \pm 0.161$ and $2.298 \pm 0.892$ respectively. We compare our model to MMDiff (Morehead et al., 2023), a protein-RNA-DNA complex co-design diffusion model. As the original best-performing version of MMDiff was trained on shorter RNA sequences, we retrain it on our training set. We also inverse-fold MMDiff's backbones using gRNAde.

We identify $N_T = 50$ as the best-performing model that balances validity, diversity, and novelty; furthermore, it takes 4.74 seconds (averaged over 5 runs) to sample a backbone of length 100, as opposed to 27.3 seconds

Table 1: **Unconditional RNA backbone generation**. We evaluate the performance of RNA-FRAMEFLOW for multiple values for denoising steps $N_T$. The best-performing model uses $N_T = 50$ steps, taking 4.74s to sample a backbone of length 100. We include the forward-folding model used in the self-consistency pipeline. We green-highlight the best result per column.

| Model | Timesteps $N_T$ | % Validity ↑ | Diversity ↑ | Novelty ↓ |
|---|---|---|---|---|
| RNA-FRAMEFLOW$_{\text{RhoFold}}$ | 10 | 16.7 | **0.62** | 0.70 |
| | 50 | **41.0** | 0.61 | **0.54** |
| | 100 | 20.0 | 0.61 | 0.69 |
| | 500 | 20.0 | 0.57 | 0.67 |
| MMDiff | 100 | 0.0 | - | - |

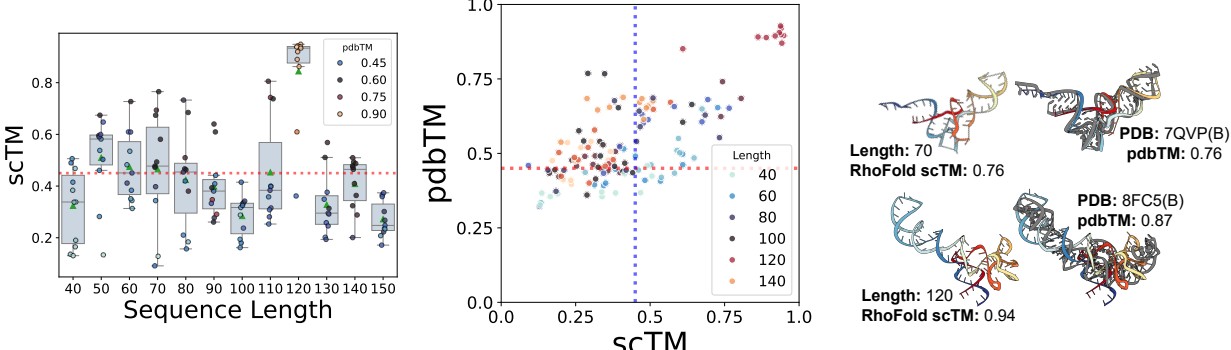

Figure 3: **Validity and novelty of generated backbones. (Left)** `scTM` of backbones of lengths 40-150 with the mean and spread of `scTM` for each length; we select the top 10 structures with the best validation scores per length. **(Middle)** Scatter plot of self-consistency TM-score (`scTM`) and novelty (`pdbTM`) across lengths. Vertical and horizontal dotted lines represent TM-score thresholds of 0.45. **(Right)** Selected samples with high pdbTM scores (colored) with the closest, aligned match from the PDB (gray). Our model generates valid backbones for certain sequence lengths and tends to recapitulate the most frequent folds in the PDB (e.g., tRNAs, small rRNAs).

for MMDiff with 100 diffusion steps. We note that increasing $N_T$ does not improve validity despite allowing the model to perform more updates to atomic coordinate placements. Our model also outperforms MMDiff. On manual inspection, samples from MMDiff had significant chain breaks and disconnected floating strands; see Appendix C.1. We report fine-grained self-consistency metrics factoring in the *rotational* component of frames as well as all backbone atoms in Appendix B.4.

## 4.2 Local evaluation with structural measurements

For our best-performing model using diffusion timesteps $N_T = 50$, we plot histograms of bond distance, bond angles, and dihedral angles in Figure 4 (Subplots 1-3). We include the Earth Mover's distance (EMD) between measurements from the training and generated distributions as an indicator of local realism (using 30 bins for each quantity). An ideal generative model will score an EMD close to 0.0 (i.e. consistent with the training set comprising naturally occurring RNA). In Table 2, we observe EMD values from our best-performing model's backbones being significantly closer to 0.0 compared to MMDiff. We include histograms of local structural descriptors for MMDiff in Appendix C.1.

We also show RNA Ramachandran angle plots for generated samples and the training distribution in Figure 4 (Subplot 4). Keating et al. (2011) introduce $\eta - \theta$ plots, similar to Ramachandran angle plots for proteins, that track the separate dihedral angles formed by $\{C4'_i, P_{i+1}, C4'_{i+1}, P_{i+2}\}$ and $\{P_i, C4'_i, P_{i+1}, C4'_{i+1}\}$ respectively, for each nucleotide $i$ along the chain. We observe that the dihedral angle distribution from RNA-FRAMEFLOW closely recapitulates the angular distribution from naturally occurring RNA structures

| Model | Earth Mover's Distance (↓) | | |
|---|---|---|---|
| | distance | angles | torsions |
| 50/50 training dist. | $6.25 \times 10^{-2}$ | $8.97 \times 10^{-4}$ | $7.24 \times 10^{-5}$ |
| RNA-FRAMEFLOW ($N_T = 50$) | **0.17** | **0.11** | **2.36** |
| MMDiff (original) | 1.38 | 0.43 | 3.06 |
| MMDiff (retrained) | 0.39 | 0.21 | 3.23 |
| Gaussian noise | 29.00 | 6.35 | 4.37 |

Table 2: **Local structural metrics.** Earth Mover's Distance for local structural measurements compared to ground truth measurements from RNAsolo. We also include EMD for a 50/50 train split as a sanity check. Our model shows improved recapitulation of local structural descriptors compared to baselines.

| Model | % Validity ↑ | Diversity ↑ | Novelty ↓ |
|---|---|---|---|
| Base | **41.0** | 0.62 | 0.54 |
| + Clustering | 12.0 | **0.88** | 0.49 |
| + Cropping | 11.0 | 0.85 | **0.47** |

Table 3: **Impact of data preparation strategies.** Increasing the diversity of the training dataset using a combination of strategies improves diversity and novelty of generated structures but leads to fewer designs passing the validity threshold.

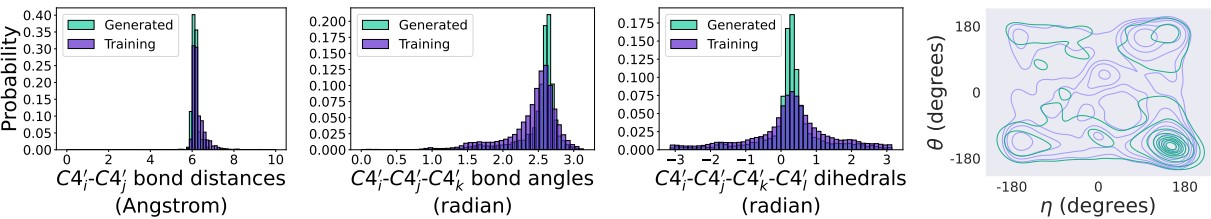

Figure 4: **Local structural metrics** from 600 generated backbone samples, compared to random Gaussian point cloud as a sanity check. Our model can recapitulate local structural descriptors. **(Subplots 1-3)** Histograms of inter-nucleotide bond distances, bond angles between nucleotide triplets, and torsion angles between every four nucleotides. **(Subplot 4)**: RNA-centric Ramachandran plot of structures from the training set (purple) and generated backbones (green).

in the training set. We provide additional evidence on the successful modeling of *ring puckering* in RNA-FRAMEFLOW's generated backbones in Appendix C.6.

### 4.3 Generation quality across sequence lengths

We next investigate how sequence length affects the global realism of generated samples (measured by `scTM`). Figure 3 (Left) shows the performance of RNA-FRAMEFLOW for different sequence lengths. We observe our model generates samples with high `scTM` for specific sequence lengths like 50, 60, 70, and 120 while generating poorer quality structures for other lengths. We believe the overrepresentation of certain lengths in the training distribution causes the fluctuation of TM-scores. We can also partially attribute this to the inherent length bias of RhoFold; see Appendix A.3. With better structure predictors, we expect more samples to be *valid*. We provide additional local evaluations of angular distributions in Appendix C.3. We also provide an ablation using Chai-1 (Boitreaud et al., 2024) as the forward-folding model in Appendix B.3; we do not observe a significant change in *validity* with Chai-1.

We also analyze the novelty of our generated samples (measured by `pdbTM`) in Figure 3 (Middle). We are particularly interested in samples that lie in the right half with high `scTM` and low `pdbTM`, which means that the designs are highly likely to fold back into the sampled backbone but are structurally dissimilar to any RNAs in the training set. It is worth noting that our training set has high structural similarity among samples: running `qTMclust` on our training dataset revealed only 342 unique clusters from 5,319 samples, which indicates that the model does not encounter a diverse set of samples during training. This contributes to many generated samples from our model looking similar to samples from the training distribution. We include two such examples in Figure 3 (Right). Both generated RNAs yield relatively high `pdbTM` scores and look similar to their respective closest matching chain from the training set: a tRNA at length 70 and a 5S ribosomal RNA at length 120, respectively. We include comparative results on validity and novelty for MMDiff in Appendix C.1, finding that MMDiff does not generate any samples that pass the validity criteria.

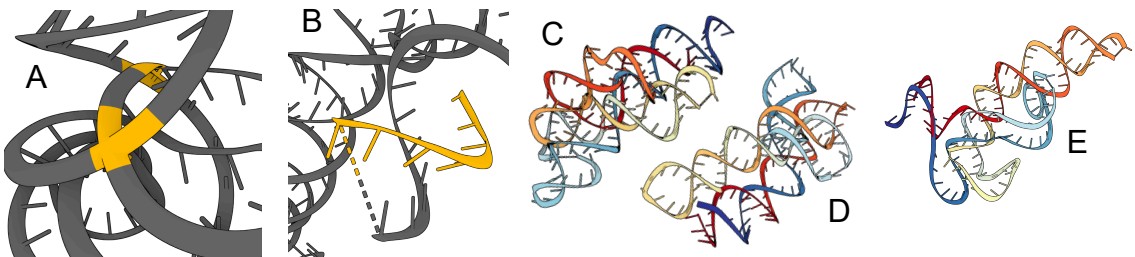

Figure 5: **Physical violations in generated samples.** (A) Inter-chain clashes (highlighted yellow). (B) Chain breaks and stray strands (highlighted yellow). (C)-(E) Excessive loops and helices.

### 4.4 Data preparation protocols

Due to the overrepresentation of RNA strands of certain lengths (mostly corresponding to tRNA or 5S ribosomal RNA) in our training set, our models generate close likenesses for those lengths that achieve high self-consistency but are not novel folds. To avoid this memorized recapitulation and promote increased diversity among samples, we sought to develop data preparation protocols to balance RNA folds across sequence lengths. We identically train RNA-FRAMEFLOW on these data splits for 120K gradient steps, with results reported in Table 3.

- **Structural clustering:** We cluster our training set using `qTMclust`. When creating a training batch, we sample random clusters, and from each cluster, random structures. This ensures batches do not solely contain samples for a single sequence length or are dominated by over-represented folds. There are only 342 structural clusters for the 5,319 samples within sequence lengths 40-150, highlighting the lack of diversity in RNA structural data. Each batch comprises padded samples up to a maximum length of 150 from randomly selected clusters across sequence lengths.

- **Cropping augmentation:** We expand our training set by cropping longer RNA strands beyond length 150 by sampling a random crop length in $[40, 150]$ and extracting a contiguous segment from the larger chains. As cropped RNA are not standalone molecules and serve only to augment the dataset, we consider a randomly chosen 20% of the training set size to balance uncropped and cropped samples; this gives 1,063 extra cropped samples.

We observe improved diversity and novelty at the cost of reduced validity. Randomly cropping may introduce subsequences that fold into significantly different structures than the substructure extracted from the original RNA; these subsequences may even unfold in real life. As a result, the augmented dataset may contain folds that are unstable or implausible. The structure prediction and inverse-folding models may not have encountered these folds loosely recapitulated by our model, resulting in poor validity. We are actively developing principled cropping methods that capture unique, realistic folds. We include additional results on these data preparation protocols in Appendix C.2.

## 5 Limitations and Discussions

Altogether, our experiments demonstrate that the $SE(3)$ flow matching framework is sufficiently expressive for learning the distribution of 3D RNA structure and generating realistic RNA backbones similar to well-represented RNA folds in the PDB. Select examples are shown in Figure 6. We have also identified notable limitations and avenues for future work, which we highlight below.

**Physical violations.** While well-trained models usually generate realistic RNA backbones, we do observe some physical violations: generated backbones sometimes have chains that are either too close by or directly clash with one another, are highly coiled, have excessive loops and unrealistically intertwined helices, or have chain breaks. We highlight these limitations in Figure 5 and analyze steric clashes in our generated backbones in Appendix C.4. RNA tertiary structure folding is driven by *base pairing* and *base stacking* which influence the formation of helices, loops, and other tertiary motifs (Vicens & Kieft, 2022). Base pairing refers to nucleotides along adjacent chains forming hydrogen bonds, while base stacking involves interactions between

rings of adjacent nucleotide bases along a chain. RNA-FRAMEFLOW attempts to model these phenomena using auxiliary losses but still operates at the backbone-only granularity, not including base atoms (barring $N1/N9$). An array of complex tertiary folds (e.g., kissing loops, tetraloops, junctions) occur through base-mediated interactions, meaning RNA-FRAMEFLOW can only implicitly learn base pairing and stacking. Developing explicit representations of these interactions as part of the architecture may further minimize physical violations and provide stronger inductive biases to learn complex tertiary RNA motifs.

**Generalization and novelty.** We observed that the best designs from our models (as measured by scTM score) are sampled at lengths 70-80 and 120-130, and often have closely matching structures in the PDB (high TM-scores). This suggests that models can recapitulate well-represented RNA folds in their training distribution (e.g., both tRNAs at length 70-90 and small 5S ribosomal RNAs at length 120 are very frequent). However, self-consistency metrics were relatively poorer for less frequent lengths, suggesting that models are currently not designing novel folds.

We would also like to note that the models we use for structure prediction and inverse folding may be similarly biased to perform well for certain sequence lengths, leading to the overall pipeline being reliable for commonly occurring lengths and unreliable for less frequent ones. We also provide an empirical upper bound on the error accumulated in the self-consistency pipeline in Appendix A.2. We evaluated preliminary strategies for structural clustering and cropping augmentations during training, which improved the novelty of designed structures but led to fewer designs passing the validity filter. Overall, the relative scarcity of RNA structural data compared to proteins necessitates greater care in preparing data pipelines for scaling up training and incorporating inductive biases into generative models.

## 6  Related Work

Recent end-to-end RNA structure prediction tools include RhoFold (Shen et al., 2022), RoseTTAFold2NA (Baek et al., 2022a), DRFold (Li et al., 2023a), and AlphaFold3 (Abramson et al., 2024), each with varying performance that is yet to match the current state-of-the-art for proteins. Other approaches use GNNs as ranking functions (Townshend et al., 2021) together with sampling algorithms (Boniecki et al., 2016; Watkins et al., 2020). However, structure prediction tools are not directly capable of designing new structures, which this work aims to address by adapting an $SE(3)$ flow matching framework for proteins (Yim et al., 2023a). MMDiff (Morehead et al., 2023), a diffusion model for protein-nucleic acid complex generation, can also sample RNA-only structures in principle. Our evaluation shows that our flow matching model significantly outperforms both the original and RNA-only versions of MMDiff that we re-trained for fair comparison.

Joshi et al. (2025) introduce gRNAde, a GNN-based encoder-decoder for 3D RNA inverse folding, a closely related task of designing new sequences conditioned on backbone structures. Tan et al. (2023) and Shulgina et al. (2024) have also developed GNNs for 3D RNA inverse folding. We use gRNAde (Joshi et al., 2025) followed by RhoFold (Shen et al., 2022) in our evaluation pipeline to forward fold designed backbones and measure structural self-consistency.

Independently and concurrent to our work, Nori & Jin (2024) propose RNAFlow, an SE(3) flow matching model to co-design RNA sequence and structure conditioned on protein partners. At each denoising step, RNAFlow uses a protein-conditioned variant of gRNAde (Joshi et al., 2025) to inverse fold noised structures, followed by RoseTTAFold2NA (Baek et al., 2022a) to predict the structure of the designed sequence. The performance of RNAFlow is upper-bounded by RoseTTAFold2NA as a pre-trained structure generator, which is kept frozen and not developed for designed RNAs which do not have co-evolutionary MSA information. Our work tackles *de novo* 3D RNA backbone generation, an orthogonal design task of sampling RNA backbone structures. We train RNA structure generation models from scratch, akin to recent developments in protein design (Yim et al., 2023b;a; Bose et al., 2023; Lin & AlQuraishi, 2023). Backbone generation followed by inverse folding has shown experimental success in designing functional proteins (Dauparas et al., 2022; Watson et al., 2023; Ingraham et al., 2022), as the framework is flexible for including specific structural motifs and sequence constraints. Since the original release of RNA-FRAMEFLOW and its source code in 2024, several methods employ a similar $SE(3)$ flow matching setup. Rubin et al. (2025) incorporate an additional sequence track for RNA structure-sequence co-design, using the MultiFlow framework from Campbell et al. (2024). Tarafder & Bhattacharya (2025) condition the 3D structure generation on pre-determined sequences and secondary structural contact maps, enabling all-atom resolution with base identities.

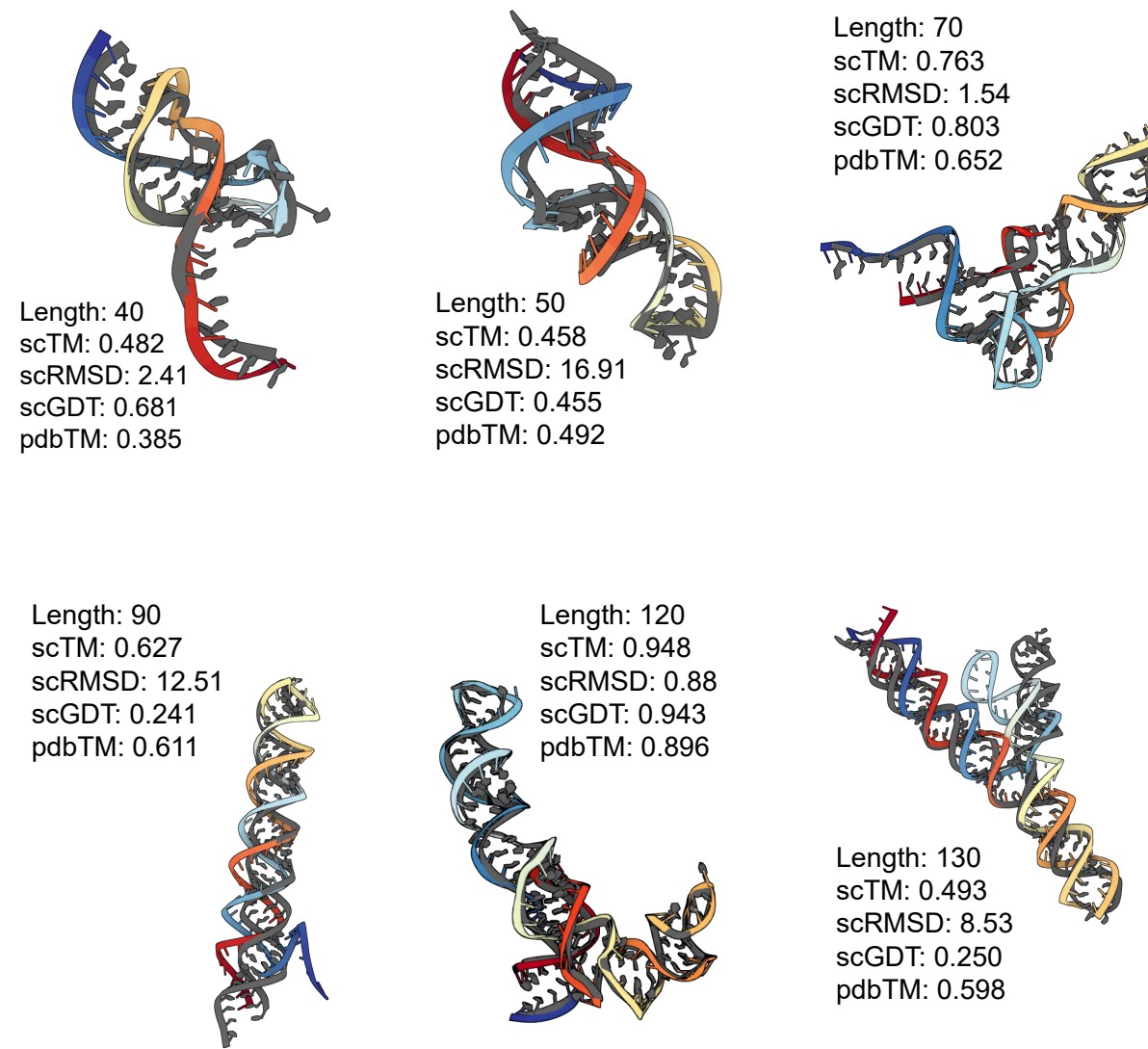

Figure 6: **Generated RNA backbones** (colored) of varying lengths aligned with their RhoFold-predicted structure (gray). We provide post-evaluation metadata obtained from our self-consistency pipeline.

## 7 Conclusion

We introduce RNA-FRAMEFLOW, a generative model for 3D RNA backbone design. Our evaluations show that our model can design locally realistic and moderately novel backbones of length $40 - 150$ nucleotides. We achieve a validity score of 41.0% and relatively strong diversity and novelty scores compared to diffusion model baselines and ablated variants. While generative models can successfully recapitulate well-represented RNA folds (e.g., tRNAs, small rRNAs), the lack of diversity in the training data may hinder broad generalization at present. Directions for future research include exploring improved data preparation strategies combined with inductive biases that explicitly incorporate physical interactions that drive RNA structure, including base pairing and base stacking. We hope RNA-FRAMEFLOW and the associated evaluation framework can serve as foundations for the community to explore 3D RNA design, towards developing conditional generative models for real-world design scenarios.

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

# Appendices

# A Additional Experimental Details

## A.1 Denoiser Hyperparameters

| Category | Hyperparameter | Value |
|---|---|---|
| Invariant Point Attention (IPA) | Atom embedding dimension $D_h$ | 256 |
| | Hidden dimension $D_z$ | 128 |
| | Number of blocks | 6 |
| | Query and key points | 8 |
| | Number of heads | 8 |
| | Key points | 12 |
| Transformer | Number of heads | 4 |
| | Number of layers | 2 |
| Torsion Prediction MLP | Input dimension | 256 |
| | Hidden dimension | 128 |
| Schedule | Translations (training / sampling) | linear / linear |
| | Rotations (training / sampling) | linear / exponential |
| | Number of denoising steps $N_T$ | 50 |

Table 4: Hyperparameters for best performing denoiser model.

## A.2 Upper bound performance of the Self-consistency Pipeline

Our self-consistency pipeline to compute *validity* involves inverse and forward folding using gRNAde (Joshi et al., 2025) and RhoFold (Shen et al., 2022). Placing upper bounds on the performance of our RNA backbone design pipeline offers insights into areas of improvement using available open-source tools.

To quantify the total error accumulated in our self-consistency pipeline, and its impact on downstream *validity*, we study the extent to which gRNAde and RhoFold can retrieve the ground truth sequences and structures from the RNAsolo training set. To assess RhoFold's structure prediction performance, we take all ground truth sequences of length $40 - 150$ from RNAsolo, forward-fold (FF) them using RhoFold, and compute self-consistency metrics (TM-score, RMSD) by comparing them with the sequences' associated 3D folds. To assess gRNAde's sequence recovery performance, we inverse-fold (IF) 3D backbones from RNAsolo through gRNAde to get 16 likely sequences and pass them to RhoFold for forward-folding.

As shown in the table below, the average self-consistency of the gRNAde-RhoFold pipeline with RNAsolo ground truth backbone structures is 43.7%, close to RNA-FRAMEFLOW's *validity* of 41.0%. This shows us that the generated backbones from RNA-FRAMEFLOW closely retain the validity of RNAsolo backbones and corresponding sequences from gRNAde. In Figure 7, we also show the self-consistency TM-scores per length bins.

| Pipeline | Self-consistency (%) ↑ | Avg `scTM` ↑ | Avg `scRMSD` ↓ |
|---|---|---|---|
| RNAsolo + FF only | 55.1 | 0.690 | 2.804 |
| RNAsolo + IF + FF | 43.7 | 0.663 | 3.085 |
| RNA-FRAMEFLOW + IF + FF (ours) | 41.0 | 0.641 | 2.298 |

## A.3 RhoFold Length Bias

We investigate the performance of RhoFold on a representative subset of the training dataset used to train RNA-FRAMEFLOW. Figure 8 shows that RhoFold has a sequence length bias where it predicts accurate structures with low RMSDs (to the ground truth) for specific sequence lengths (like 70, 100, and 120) while predicting poor structures for other lengths. The performance across lengths is disparate and may influence what is considered *valid* in our unconditional generation benchmarks. This affects its efficacy when used in a self-consistency pipeline with the RMSD metric. To minimize the influence of this length bias, we use TM-score for self-consistency because it does not penalize flexible regions as much as RMSD.

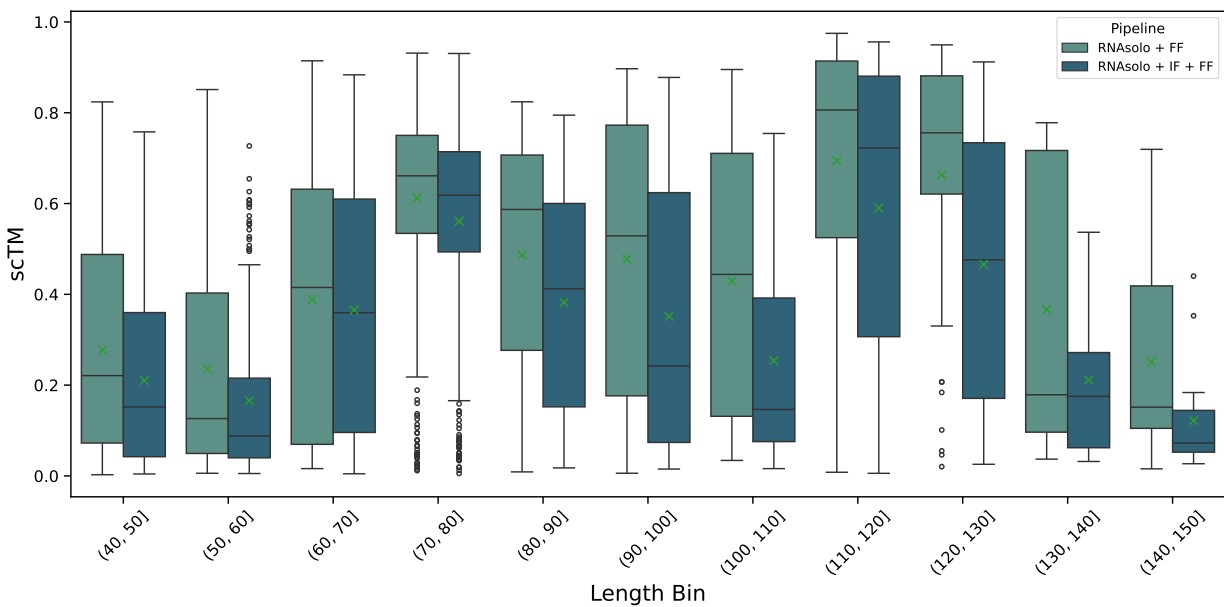

Figure 7: **Self-consistency scores on RNAsolo samples by sequence length**. We observe that generated backbones from RNA-FRAMEFLOW retain the self-consistency of gRNAde-predicted sequences.

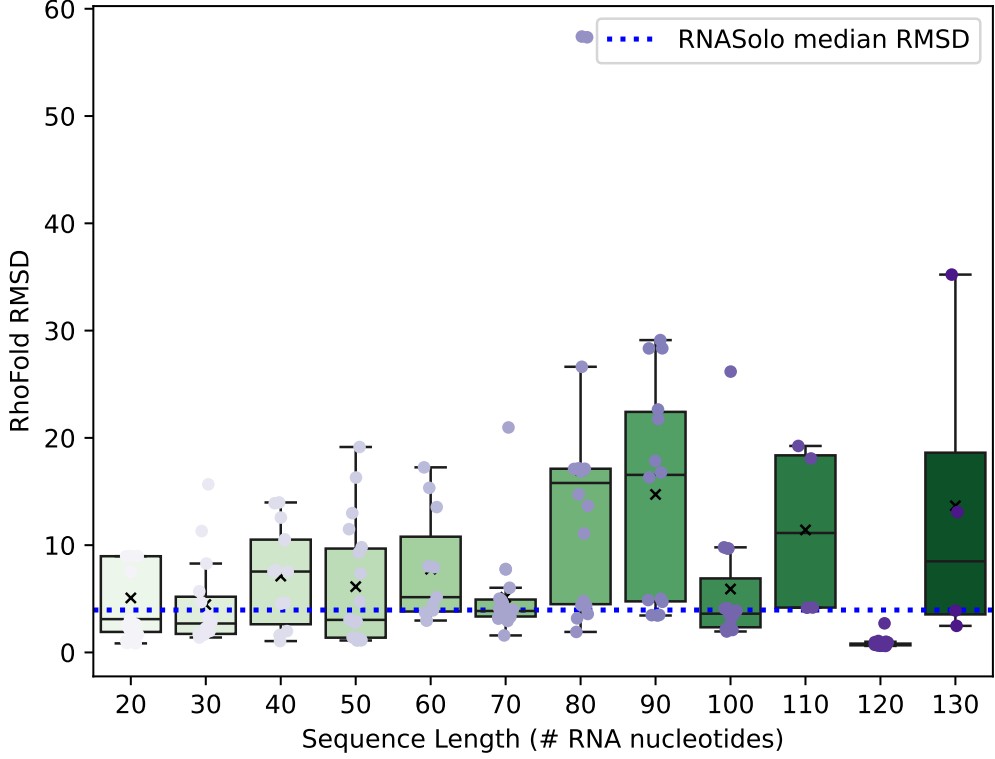

Figure 8: **RhoFold length bias**. The blue dotted line represents the median RMSD of RhoFold predictions to the RNAsolo samples. RhoFold performs well for over-represented sequence lengths in the PDB, and poorly for under-represented sequence lengths.

### A.4 Imputing Non-frame Atoms from Torsion Angles

Here, we describe how we autoregressively impute the remaining non-frame atoms using 8 torsion angles $\Phi = \{\phi_1 \rightarrow \phi_8\}$. For a nucleotide $n$ along the generated RNA backbone, we assume we have its final frame $T^{(n)} = (r^{(n)}, x^{(n)})$ obtained from the denoiser's output after $N_T$ diffusion timesteps. Going by our choice of frame $\{C3', C4', O4'\}$, we place non-frame atoms in the following order: $C2', C1', N1/N9, O3', O5', P, OP1, OP2$, each corresponding to its respective $\phi_i \in \Phi$ as shown in Figure 1.

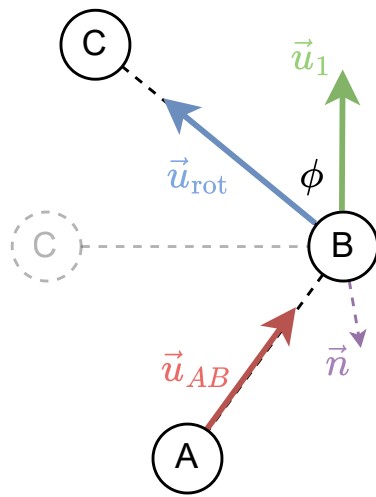

Referring to the figure on the right, suppose we have three atoms A, B, and C with coordinates $(x_A, y_A, z_A), (x_B, y_B, z_B), (x_C, y_C, z_C)$. They are connected by bonds AB and BC denoted by vectors $\overrightarrow{AB} = B - A$ and $\overrightarrow{BC} = C - B$ with lengths $r_{AB} = |\overrightarrow{AB}|$ and $r_{BC} = |\overrightarrow{BC}|$ respectively. To rotate $BC$ around $AB$ by some angle $\phi$, we perform the following procedure:

1. Compute unit vector $\vec{u}_{AB} = \frac{\overrightarrow{AB}}{r_{AB}}$ along bond AB by normalizing $\overrightarrow{AB}$.

2. Compute a vector perpendicular to $\vec{u}_{AB}$ by choosing a random normal vector $\vec{n}$ (like $[1, 0, 0]^T$ or $[0, 1, 0]^T$) and taking their cross product to get $\vec{u}_1 = \vec{u}_{AB} \times \vec{n}$.

3. Compute the unit vector $\vec{u}_{\text{rot}}$ rotated by $\phi$ around $AB$ using Rodrigues' rotation formula:

$$\vec{u}_{\text{rot}} = \cos{(\phi)} \cdot \vec{u}_1 + \sin{(\phi)} \cdot (\vec{u}_{AB} \times \vec{u}_1) + (1 - \cos{(\phi)})(\vec{u}_{AB} \cdot \vec{u}_1) \cdot \vec{u}_{AB} \ .$$

4. Compute the coordinates of atom $C$ as follows:

$$[x_C, y_C, z_C] = [x_B, y_B, z_B] + r_{BC} \cdot \vec{u}_{\text{rot}} \ .$$

We use predetermined bond lengths between atoms in the idealized geometry of the Adenine (A) nucleotide from OpenComplex (Jingcheng et al., 2022) in the same way Yim et al. (2023b;a) use Alanine for generated protein backbones. We use the following atom triplets and predicted torsion angles to build the all-atom nucleotide, starting from the ribose sugar ring towards the $5'$ end (i.e., the phosphate group):

| Fixed bond | Non-frame atom | Torsion angle |
|:---:|:---:|:---:|
| $C4' - C3'$ | $C2'$ | $\phi_1$ |
| $C4' - O4'$ | $C1'$ | $\phi_2$ |
| $O4' - C1'$ | $N9$ (or $N1$) | $\phi_3$ |
| $C4' - C3'$ | $O3'$ | $\phi_4$ |
| $C4' - C5'$ | $O5'$ | $\phi_5$ |
| $C5' - O5'$ | $P$ | $\phi_6$ |
| $O5' - P$ | $OP1$ | $\phi_7$ |
| $O5' - P$ | $OP2$ | $\phi_8$ |

# B   Ablations

## B.1   Composition of Backbone Coordinate Loss

We also analyze how changing the composition of atoms in the inter-atom losses affects performance. We increase the number of atoms being supervised in the $\mathcal{L}_{\text{bb}}$ loss described above. Aside from the frame comprising $\{C3', C4', O4'\}$, we try two settings with 3 and 7 additional non-frame atoms included in the loss. For the 3 non-frame atoms, we additionally choose $\{C1', P, O3'\}$, and for the 7 non-frame atoms, we choose a superset $\{C1', P, O3', C5', OP1, OP2, N1/N9\}$. We posit the additional supervision may increase the local structural realism, which may further improve validity, as shown in Table 5.

We indeed observe increasing validity as we increase the frame complexity in the auxiliary backbone loss. The minute RMSD contributions from disordered fragments of the RNA may be minimal, accounting for greater likeness to the RhoFold predicted structures, scoring relatively higher scTM scores. However, the original frame-only baseline model has better diversity and novelty which we attribute to high local variation in atomic placements. This variation causes two generated structures for the same sequence length to look very different at an all-atom resolution.

| Frame composition in $\mathcal{L}_{\text{bb}}$ | % Validity ↑ | Diversity ↑ | Novelty ↓ |
|---|---|---|---|
| Frame only (baseline) | 41.0 | **0.62** | **0.54** |
| Frame and 3 non-frame | 45.0 | 0.28 | 0.79 |
| Frame and 7 non-frame | **46.7** | 0.35 | 0.85 |

Table 5: Ablating composition of backbone loss $\mathcal{L}_{\text{bb}}$. Supervising more non-frame atoms improves validity but worsens diversity and novelty. Best result per column is highlighted.

## B.2   Composition of Auxiliary Loss

We ablate the inclusion of different auxiliary loss terms that guide our $SE(3)$ flow matching setup; results are in Table 6. Although, there is an increase in EMD for bond distances as we remove distance-based losses like backbone coordinate loss $\mathcal{L}_{\text{bb}}$ and all-to-all pairwise distance loss ($\mathcal{L}_{\text{dist}}$). However, we also observe the model still learns realistic distributions despite removing different loss terms, indicating that each loss makes up for the absence of the other. Moreover, the best model still uses all losses with any removal causing a drop in validity. Further inspecting the samples from the models without each loss term reveals structural deformities at the all-atom level. Figure 9 shows such artifacts resulting from not enforcing geometric constraints through explicit losses.

| $\mathcal{L}_{\text{bb}}$ | $\mathcal{L}_{\text{dist}}$ | $\mathcal{L}_{SO(3)}$ | EMD (distance) ↓ | EMD (angles) ↓ | EMD (torsions) ↓ | % Validity ↑ |
|---|---|---|---|---|---|---|
| ✓ | ✓ | ✓ | **0.17** | **0.11** | **2.36** | **41.0** |
| ✓ |   | ✓ | 0.18 | 0.14 | 3.85 | 35.0 |
| ✓ | ✓ |   | 0.23 | 0.11 | 3.72 | 13.3 |
|   | ✓ | ✓ | 0.18 | 0.18 | 3.59 | 16.7 |

Table 6: Ablations of loss terms on Earth Mover's Distance scores for structural measurements compared to ground truth measurements from the training set. The first row corresponds to the baseline model. Distance-based losses like the backbone coordinate loss ($\mathcal{L}_{\text{bb}}$) and all-to-all pairwise distance loss ($\mathcal{L}_{\text{dist}}$) are necessary to learn geometric properties like bond distances adequately.

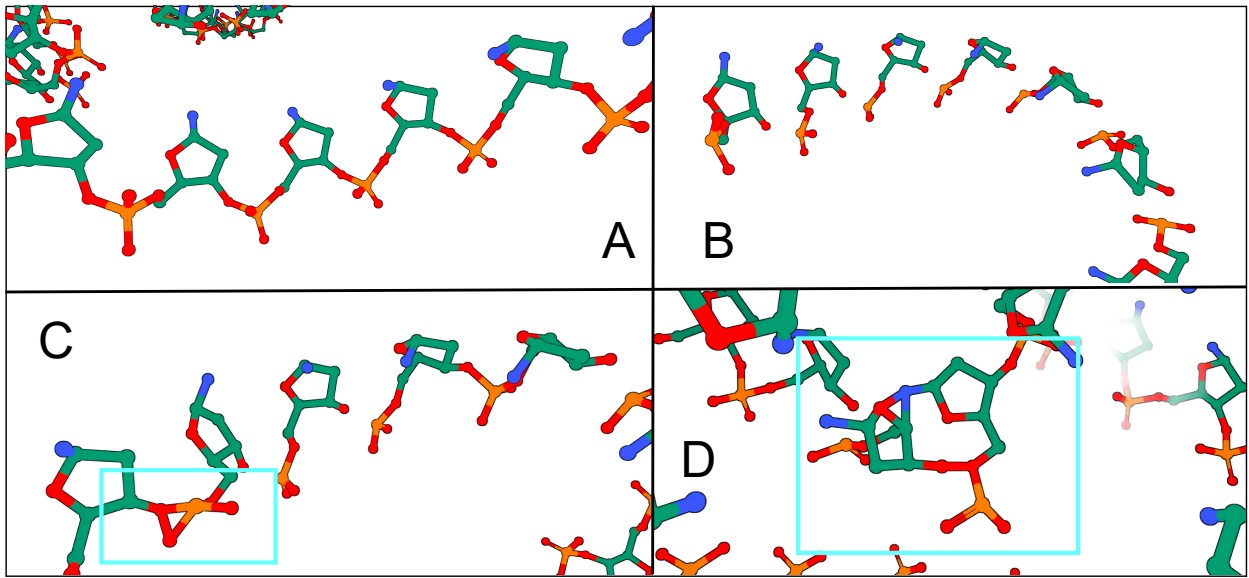

Figure 9: Not including auxiliary losses causes structural deformities in generated RNAs. **(A)** RNA backbone from our baseline model with expected adherence to bonding between nucleotides. **(B)** Not including the rotation loss $\mathcal{L}_{SO(3)}$ causes nucleotides to have random orientations, preventing them from connecting contiguously. **(C)** Not including the backbone atom loss $\mathcal{L}_{\text{bb}}$ places intra-residue atoms too close to one another resulting in bonds that should not exist. **(D)** Not including the all-to-all pairwise distance loss $\mathcal{L}_{\text{dist}}$ causes adjacent frames to fuse and loses contiguity, especially along helices and loops.

### B.3 Choice of Forward-folding Model

In our work, we rely on RhoFold (Shen et al., 2022) to forward fold the inverse-folded sequences from gRNAde. Here, we reperform our evaluation from Section 4.1 with Chai-1 (Boitreaud et al., 2024), a recent open-source structure prediction model with results similar to AlphaFold2, replacing RhoFold in the self-consistency pipeline in Figure 2. We do not use MSAs for Chai-1. We do not observe any significant difference in self-consistency distributions: for RNA-FRAMEFLOW_RhoFold, we report a *validity* of 41.0% while RNA-FRAMEFLOW_Chai-1 gives a *validity* of 39.5%.

Recent benchmarks (Tarafder et al., 2024) also observe that existing RNA structure prediction tools like RhoFold, RF2NA (Baek et al., 2022b), and tr-RosettaRNA (Wang et al., 2023) perform similarly due to similarities in their architectures and training data. For 600 generated backbones from RNA-FRAMEFLOW, we get 8 predicted sequences from gRNAde, giving us 4800 predicted backbones from RhoFold and Chai-1. We compare this scTM distribution across predicted structures in Figure 10.

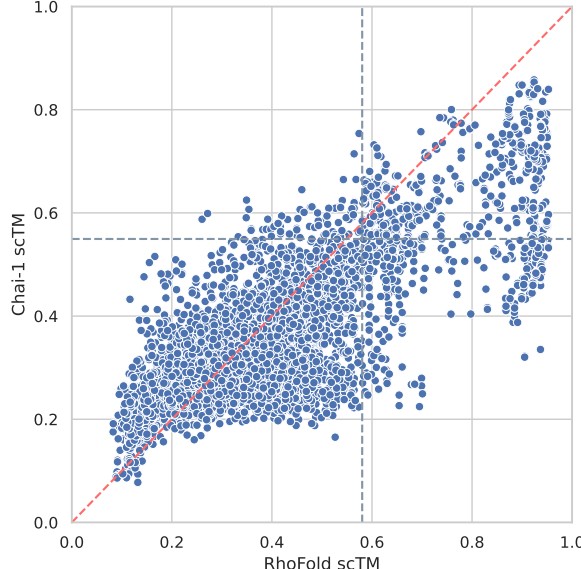

Figure 10: **Correlation between RhoFold and Chai-1 scTM scores.** Horizontal and vertical dotted lines denote the median scTM score from each method across all samples.

### B.4 Rotational and All-atom Self-consistency

Our self-consistency pipeline currently computes TM-score and RMSD with $C4'$ coarse graining, reflecting performance only along the translational component. However, our frames not only comprise the $C4'$ atom but the $C3'$ and $O4'$ atoms as well, forming a rotational component. Factoring in this *rotational* component into the self-consistency metrics would offer a clearer picture of RNA-FRAMEFLOW's ability to precisely orient frames. In Table 7, we report `scRMSD` statistics over our generated backbones. We observe that the `scRMSD` values from RNA-FRAMEFLOW samples correlate positively with the self-consistency scores in Figure 3 (left). For sequence lengths with high % *validity*, we see relatively lower `scRMSD` values and higher `scTM` scores.

| Sequence length | Median `scRMSD` ↓ | Mean `scRMSD` ↓ | Std. Dev. `scRMSD` ↓ |
|---|---|---|---|
| 40 | 3.81 | 6.01 | 4.15 |
| 50 | 3.66 | 9.36 | 10.6 |
| 60 | 5.12 | 9.36 | 9.98 |
| 70 | 3.74 | 6.88 | 8.85 |
| 80 | 6.06 | 8.43 | 5.95 |
| 90 | 10.38 | 11.71 | 6.74 |
| 100 | 13.27 | 13.54 | 6.65 |
| 110 | 14.36 | 13.33 | 6.43 |
| 120 | 1.89 | 3.034 | 3.89 |
| 130 | 17.45 | 16.23 | 5.82 |
| 140 | 11.17 | 13.06 | 4.90 |
| 150 | 20.28 | 20.29 | 4.67 |

Table 7: Statistics of rotational `scRMSD` across sequence lengths. We observe RNA-FRAMEFLOW orients frames realistically, correlating positively with reported `scTM` in Section 4.1 across sequence lengths.

RNA-FRAMEFLOW generates backbones at an all-backbone-atom granularity. We additionally compute `scRMSD` across all 13 nucleotide atoms in Table 8. We similarly observe RNA-FRAMEFLOW can generate realistic fine-grained nucleotides: for sequence lengths with relatively higher % *validity*, we see lower all-atom `scRMSD` values.

| Sequence length | Median `scRMSD` ↓ | Mean `scRMSD` ↓ | Std. Dev. `scRMSD` ↓ |
|---|---|---|---|
| 40 | 4.22 | 6.36 | 3.88 |
| 50 | 4.05 | 9.73 | 10.48 |
| 60 | 5.41 | 9.65 | 9.85 |
| 70 | 4.19 | 7.27 | 8.71 |
| 80 | 6.20 | 8.70 | 5.78 |
| 90 | 10.53 | 11.87 | 6.65 |
| 100 | 13.26 | 13.65 | 6.57 |
| 110 | 14.39 | 13.48 | 6.25 |
| 120 | 2.82 | 3.85 | 3.67 |
| 130 | 17.47 | 16.31 | 5.74 |
| 140 | 11.29 | 13.19 | 4.85 |
| 150 | 20.29 | 20.32 | 4.64 |

Table 8: Statistics of all-backbone-atom `scRMSD` across sequence lengths. We observe RNA-FRAMEFLOW generates realistic fine-grained nucleotides, correlating positively with reported EMD scores for local structural descriptors in Table 2 across sequence lengths.

## C  Additional Results

### C.1  Evaluation of MMDiff Samples

Here, we document global and local metrics from samples generated by MMDiff. MMDiff has a validity score of 0.0% as all the samples have a poor `scTM` score below the 0.45 threshold to the RhoFold predicted backbones. Even though none of the samples are valid, we show the average `pdbTM` scores for the samples, which are trivially low as there are no structures from the PDB that match them due to poor quality.

While MMDiff's samples locally resemble RNA structures given realistic, manual inspection reveals multiple chain breaks and disconnected floating strands, resulting in 0.0% validity. In Figure 12 (Subplot 1), we see inter-residue $C4'$ distances slightly varying, causing the chain breaks and clashes. Furthermore, the Ramachandran plot in Figure 12 (Subplot 4) reveals a more complex angular distribution than found in the training set, which may be a consequence of excessively folded regions or substructures that may have folded in on themselves.

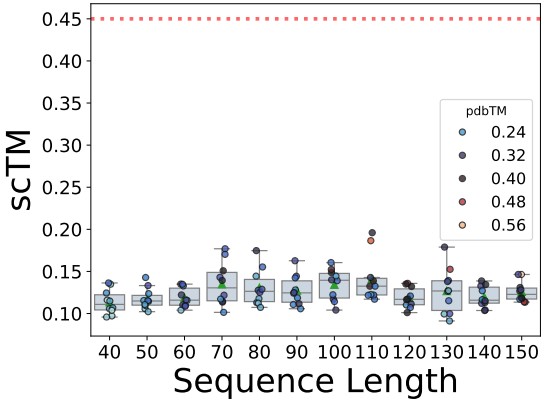
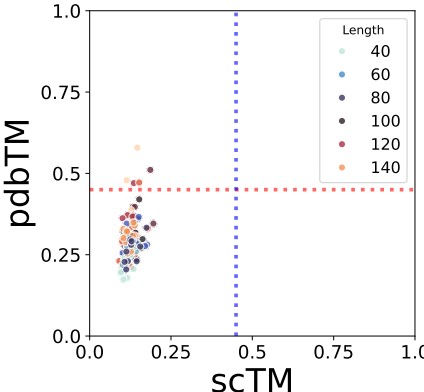

Figure 11: Validity and novelty of retrained MMDiff's top-10 generated backbones. **(Left)** `scTM` of backbones of lengths 40-150 with the mean and spread of `scTM` for each length. **(Middle)** Scatter plot of self-consistency TM-score (`scTM`) and novelty (`pdbTM`) across lengths. Vertical and horizontal dotted lines represent TM-score thresholds of 0.45. Overall, MMDiff retrained on our training set does not generate realistic RNA structures.

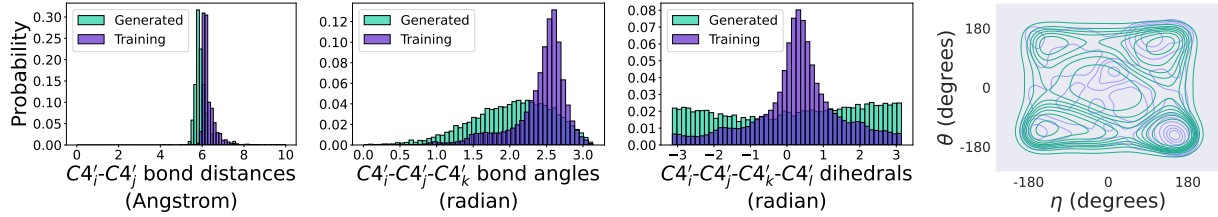

Figure 12: **Structural measurements** from samples generated by MMDiff. **(Subplots 1-3)** Left: histogram of inter-nucleotide bond distances in Angstrom. Middle: histogram of bond angles between nucleotide triplets. Right: histogram of torsion (dihedral) angles between every four nucleotides. **(Subplot 4)**: RNA-centric Ramachandran plot of structures from the training set (purple) and MMDiff's generated backbones (green).

## C.2 Evaluation of Data Preparation Strategies

We include global evaluation metrics for the two data preparation strategies presented in the main text, namely structural clustering and cropping augmentation.

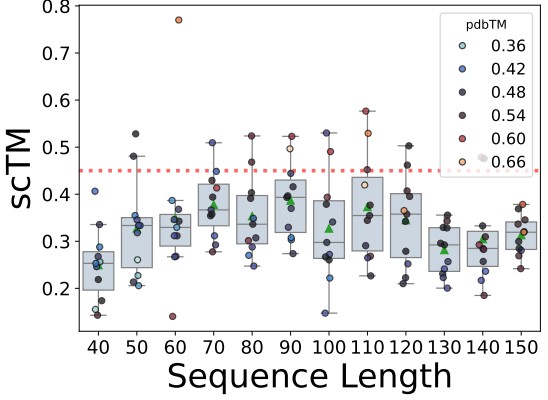 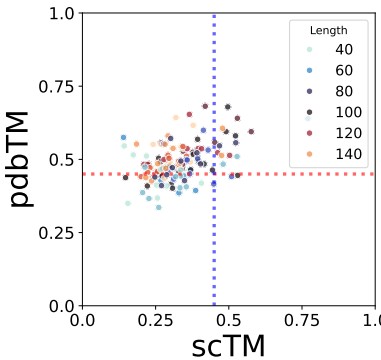

Figure 13: Validity and novelty of top-10 generated backbones from the model trained with only structural clustering. **(Left)** `scTM` of backbones of lengths 40-150 with the mean and spread of `scTM` for each length. **(Middle)** Scatter plot of self-consistency TM-score (`scTM`) and novelty (`pdbTM`) across lengths. Vertical and horizontal dotted lines represent TM-score thresholds of 0.45.

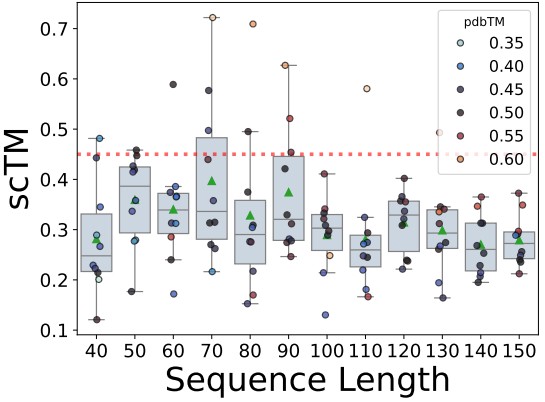 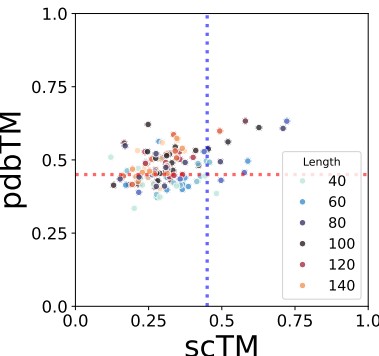

Figure 14: Validity and novelty of top-10 generated backbones from the model trained with structural clustering and cropping. **(Left)** `scTM` of backbones of lengths 40-150 with the mean and spread of `scTM` for each length. **(Middle)** Scatter plot of self-consistency TM-score (`scTM`) and novelty (`pdbTM`) across lengths. Vertical and horizontal dotted lines represent TM-score thresholds of 0.45.

### C.3 Comprehensive local evaluation of angular distributions

Following the empirical structural analysis of RNA by Gelbin et al. (1996), we compare local bond angle distributions among triplets of atoms from the generated backbones. We sample 50 all-atom backbones for each sequence length in $[70, 90, 110, 130, 150]$, sieve out the *valid* samples, and extract relevant bond angles. As shown in Figure 15, we observe that RNA-FRAMEFLOW can retrieve angular distributions between distant and nearby atoms in the nucleotides, providing preliminary evidence that modern protein design models are sufficiently expressive to model RNA tertiary structure.

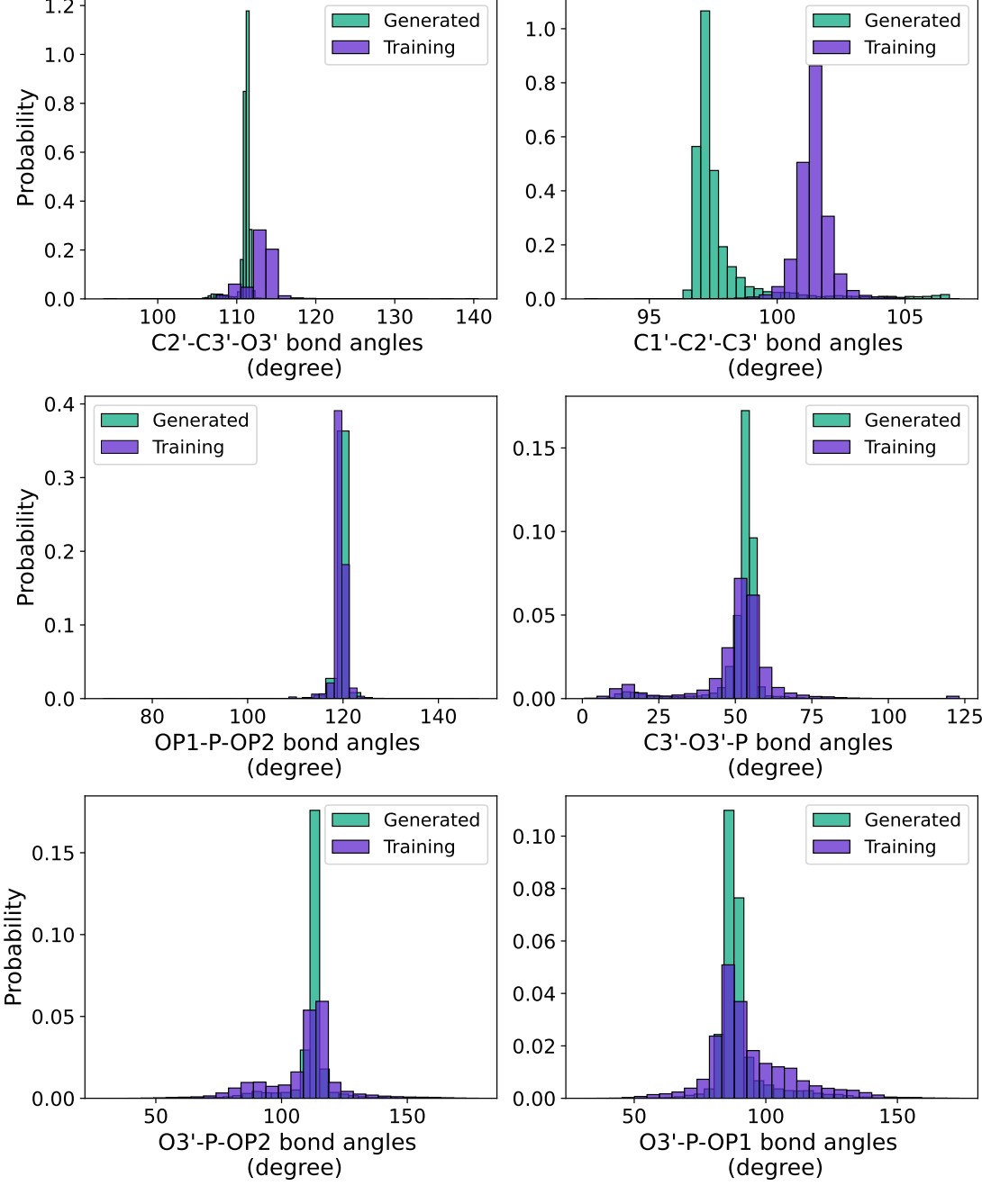

Figure 15: **Bond angle distributions between triplets of atoms**. We select these atomic triplets from the empirical study of RNA's 3D geometry by Gelbin et al. (1996).

## C.4   Measuring All-atom Steric Clashes

We compare the *all-atom-level* steric clashes between filtered RNAsolo samples used for training and the generated backbones from RNA-FRAMEFLOW. We say two *unbonded* atoms $i, j$ clash if the distance between them $r_{ij}$ is within a threshold $d_{\text{steric}}$:

$$d_{\text{steric}} = \text{v}_i + \text{v}_j - 0.6 \tag{11}$$

$$\mathbb{I}_{ij} = \begin{cases} 1 & r_{ij} \leq d_{\text{steric}} \\ 0 & \text{otherwise} \end{cases} \tag{12}$$

$$\# \text{ clashes } = \sum_{i,j} \mathbb{I}_{ij} \; . \tag{13}$$

Here, $\text{v}_i, \text{v}_j \in \mathbb{R}$ are the Van der Waals (VdW) radius of the atoms $i, j$ in Angstrom. Based on its identity, each atom has its own VdW radius which we factor into our computation. We leave a generous tolerance of 0.6 Å (corresponding to half the Hydrogen atom's VdW radius of 1.20 Å) to account for random deviations in atomic placements. We ignore Phosphodiester and Glycosidic bonds when computing clashes because the covalent radius is smaller than the VdW radius. As nucleotides are constructed using idealized bonds, there may be fewer inter-nucleotide clashes, resulting in fewer clashes for RNA-FRAMEFLOW backbones.

In Figure 16, we compare the steric clashes across sequence length bins. We observe that RNA-FRAMEFLOW generates backbones that have a similar distribution of inter-atom steric clashes as samples from RNAsolo. We also include *validity* for each sequence length bucket. We see that samples from certain sequence lengths (like 70, 80, 120) contain relatively fewer steric clashes across samples within that length bucket since they are over-represented in RNAsolo. This means RNA-FRAMEFLOW might be better at recapitulating atomic positions for such lengths than others. The steric clashes are normalized by the number of heavy atoms in the molecules, giving us steric clashes per 100 atoms. For the RNAsolo samples, we see $10.03 \pm 1.52$ clashes per 100 atoms while our generated backbones have $25.55 \pm 5.43$ clashes per 100 atoms.

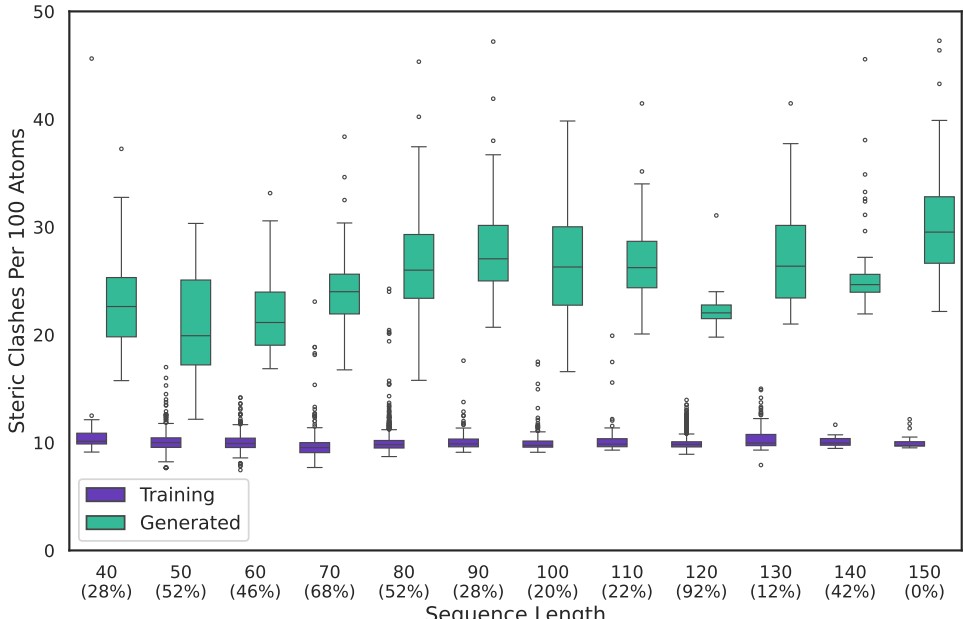

Figure 16: **All-atom steric clashes by sequence length**. We observe a similar number of steric clashes between training and generated backbones across sequence lengths. We include the (% *validity*) for generated samples from each sequence length below the length labels along the horizontal axis.

### C.5 Atomic Displacement of Frame Atoms

To further motivate our choice of frame atoms, we examine the B-factor of each atom provided in the RNAsolo PDB files. B-factor is a measure of atomic displacement, i.e., how much an atom *wiggles* in its place when the structure is undergoing X-ray crystallography or Cryo-EM. Empirically, we observe our frame construction $\{C4', C3', O4'\}$ collectively deviates the least compared to the frame construction in RF2NA (Baek et al., 2022b), $\{P, OP1, OP2\}$. We report this in Table 9 below:

| Atom | Mean B-factor ($\text{Å}^2$) ↓ | Median B-factor ($\text{Å}^2$) ↓ |
|------|------|------|
| C4' | 111.44 | 84.84 |
| C3' | 111.74 | 85.41 |
| O4' | 111.22 | 84.86 |
| P | 114.03 | 88.69 |
| OP1 | 113.46 | 87.56 |
| OP2 | 112.35 | 86.11 |

Table 9: Statistics of B-factor values (atomic displacement) of nucleotides from 500 random samples in RNAsolo. We observe our frame construction $\{C4', C3', O4'\}$ experiences lower collective spatial uncertainty compared to alternate frame constructions such as $\{P, OP1, OP2\}$.

### C.6 Modeling Ring Puckering

Puckering refers to non-planar deformations or contortions of the ribose sugar ring in RNA comprising the atoms $\{C1', C2', C3', C4', O4'\}$. Instead of forming an ideal flat plane, the ring atoms alternately move above and below the ideal plane to relieve steric strain (see Figure 17 **(A)**). The five atoms collectively define five out-of-plane torsion angles: $\nu_0$ ($C4' - O4' - C1' - C2'$), $\nu_1$ ($O4' - C1' - C2' - C3'$), $\nu_2$ ($C1' - C2' - C3' - C4'$), $\nu_3$ ($C2' - C3' - C4' - O4'$) and $\nu_4$ ($C3' - C4' - O4' - C1'$). For a nucleotide along the backbone, these angles define *Cremer–Pople pseudo-rotations* parameterized by a phase angle $P$, that locates which atom is displaced towards the *endo* position (i.e., above the flat plane), and an amplitude $\tau_m$, that quantifies the maximum magnitude of displacement. Tracking these quantities provides insights into the conformational diversity and interactions of RNA. Specifically, $C2'$-endo and $C3'$-endo puckering are known to mediate these behaviours. Several riboswitches, ribozymes, and RNA-protein interfaces exploit such *endo* orientations of ribose atoms as a trigger mechanism during molecular binding (Setlik et al., 1995; Salter et al., 2006).

In naturally-ocurring RNA (e.g., RNAsolo), the phase angle $P$ distribution is sharply bimodal with a dominant $C3'$-endo near 0° and a smaller $C2'$-endo near 180°. The puckering amplitude $\tau_m$ forms a narrow band near $32° - 38°$ (0.55-0.66 rad) (Shi et al., 2020; Harp et al., 2022). In Figure 17 **(B)**, we demonstrate that RNA-FRAMEFLOW captures key ring puckering motions, with similar distributions for $P$ and $\tau_m$.

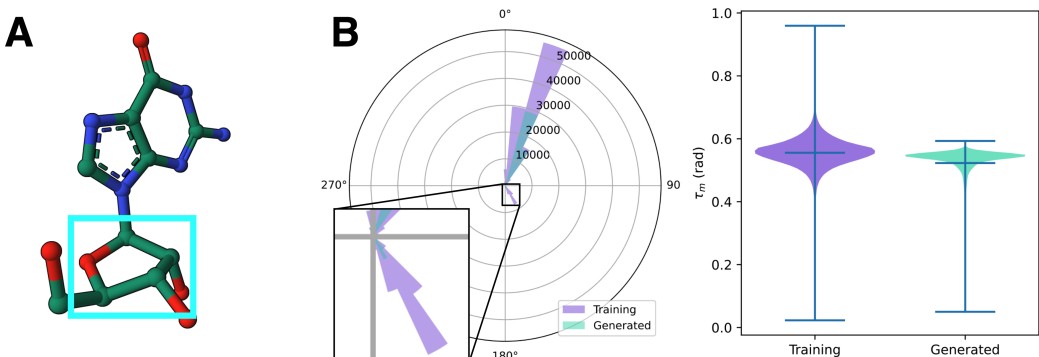

Figure 17: **Analysis of ring puckering motions. (A)** Puckering occurs when atoms comprising the ribose sugar ring move above (*endo*) or below (*exo*) the ideal flat plane (cyan box). **(B)** Rose chart of phase angle $P$ distribution (left) and violin plot of amplitude $\tau_m$ distribution (right). RNA-FRAMEFLOW (green) generates RNA backbones that capture an appropriate degree of puckering seen in natural RNA (purple).

