# OpenReview forum: "RNA-FrameFlow: Flow Matching for de novo 3D RNA Backbone Design"
_TMLR — Accepted by TMLR_

### Review · Reviewer_dYWX · 2025-05-09

**Summary Of Contributions:**

This paper introduces RNA-FrameFlow, the first generative model specifically designed for de novo 3D RNA backbone generation. The authors adapt SE(3) flow matching, a technique previously successful for protein backbone generation, to the unique challenges of RNA modeling. The key contributions include:
1. RNA Frame Parameterization: The model defines the RNA backbone frame using the rigid body formed by C4′, C3′, and O4′ atoms. The positions of the remaining backbone atoms are then determined via 8 predicted torsion angles, enabling the end-to-end generation of full-atom RNA backbones.
2. Flow Matching Training for RNA: Application of the SE(3) flow matching framework where the model learns a time-dependent vector field to map noisy, randomly initialized frames to ground-truth RNA frames. This learning process involves separate supervision of the rotation and translation components of the SE(3) transformations.
3. RNA-Specific Data Preparation and Augmentation: Development of a data preparation protocol using RNA sequences (40–150 nt) from the RNAsolo dataset. To address the scarcity and limited diversity of 3D RNA structural data, this protocol incorporates structural clustering and random cropping augmentations to expand and diversify the training samples.
4. Comprehensive Evaluation Protocol: Establishment of a rigorous evaluation suite to assess the quality of generated RNA backbones. This includes global self-consistency checks and the assessment of local structural realism.

**Audience:**

Yes

**Broader Impact Concerns:**

None. This work focuses on foundational generative modeling for RNA structures. The research is aimed at advancing scientific understanding and tool development for RNA engineering. The potential for misuse is low and in line with other generative models in computational structural biology.

**Claims And Evidence:**

Yes

**Requested Changes:**

1. Expanded Analysis of Evaluation Tool Limitations: While the appendix touches on RhoFold's biases, integrate a more prominent discussion in the main text about how the limitations of gRNAde and RhoFold might affect the interpretation of the reported "validity" and self-consistency scores. This adds a layer of critical self-assessment.
2. Expanded Analysis of Evaluation Tool Limitations: While the appendix touches on RhoFold's biases, integrate a more prominent discussion in the main text about how the limitations of gRNAde and RhoFold might affect the interpretation of the reported "validity" and self-consistency scores. This adds a layer of critical self-assessment.
3. Acknowledge Backbone-Only Limitations More Directly in Main Results: When discussing the generated structures and their novelty, more explicitly connect the backbone-only nature of the model to its potential limitations in capturing folds heavily reliant on specific base-mediated interactions. This manages reader expectations regarding the types of structures the current model can realistically generate.
4. Briefly Outline Future Directions for Conditional Generation: While a full exploration is beyond scope, a short paragraph outlining potential strategies or initial thoughts on how conditional information (e.g., secondary structure constraints) could be incorporated into the RNA-FrameFlow framework would be beneficial for readers interested in practical design applications.
5. Update Related Work/Discussion with Recent Advances: In the "Related Work" or "Discussion" sections, incorporate a brief mention and contextualization of other very recent or concurrent works in RNA backbone/structure generation (e.g., RiboGen, RNAbpFlow, or other flow/diffusion models for RNA that may have emerged). This demonstrates awareness of the rapidly evolving field.

**Strengths And Weaknesses:**

Strengths:
1. Novel Application Domain: This work is the first to adapt and apply the SE(3) flow matching paradigm to de novo RNA backbone generation, opening a promising new avenue for RNA design.
2. RNA-Specific Adaptations: The authors demonstrated significant engineering effort in adapting the SE(3) flow matching framework from proteins to RNA by thoughtfully addressing RNA's distinct structural properties; this involved defining a suitable RNA frame with a mechanism for all-atom reconstruction via predicted torsion angles, and incorporating auxiliary loss functions specifically relevant to RNA geometry.
3. Comprehensive evaluation framework: The paper establishes rigorous evaluation protocols combining global (TM-score, scRMSD) and local (bond angles, distances) metrics that provide a multi-faceted assessment of generated RNA structures.
4. Clarity and Openness: The paper is generally well-written, the methodology is clearly explained, and the authors are transparent about limitations (e.g., physical violations, novelty of folds).

Weakness:
1. Limited Technical Novelty in Core Algorithm: The fundamental flow matching mechanism is a direct adaptation of the existing FrameFlow for proteins. While the RNA-specific frame definitions and auxiliary losses are important contributions, the core generative algorithm itself does not introduce new principles to flow matching.
2. Sensitivity to Training Data Distribution and Limited Representation of RNA Conformational Space: The model's performance, particularly in achieving high scTM validity and generating truly novel folds (as measured by pdbTM), appears significantly constrained by the characteristics of the available RNA structural data, such as that in RNAsolo. Current RNA 3D structure databases are notably less extensive and diverse than their protein counterparts. Many existing RNA datasets, including RNAsolo, exhibit high structural similarity within certain sequence families and an over-representation of common classes (e.g., tRNAs, 5S rRNAs at specific lengths). This inherent lack of diversity in the training data means it may not adequately represent the full breadth of the natural RNA conformational space. Consequently, training a generative model solely on such data, even with augmentation, might inherently limit its ability to generalize and design truly diverse and novel structural motifs, potentially leading to the preferential recapitulation of observed folds rather than exploration of uncharted structural territories.
3. Inherent Uncertainty in Evaluation Pipeline: The "validity" metric is contingent upon the performance of external tools like gRNAde and RhoFold. As the authors acknowledge in the appendix, RhoFold exhibits its own length biases and inaccuracies. While this is a common challenge in evaluating generative models, it means the reported 41% validity is an estimate influenced by the capabilities and potential systematic errors of these downstream predictors, rather than an absolute measure of intrinsic structural quality.
4. Backbone-Only Representation Overlooks RNA's Primary Folding Determinants: A fundamental distinction between RNA and protein folding is their primary driving forces. While protein folding is significantly influenced by backbone hydrogen bonding and hydrophobic collapse, RNA tertiary structure is predominantly dictated by base pairing (both canonical and non-canonical) and base stacking interactions. By focusing exclusively on backbone atom generation, RNA-FrameFlow does not explicitly model base identities or these crucial base-mediated interactions. Although auxiliary losses on backbone geometry attempt to implicitly capture some consequences of these interactions, the absence of direct base modeling inherently limits the model's capacity to learn the true biophysical drivers of RNA folding. This makes it challenging to generate and stabilize complex tertiary arrangements, especially those involving intricate non-canonical interactions or long-range pairings, which are critical for RNA function.
5. Focus on Unconditional Generation: The current work primarily addresses unconditional backbone generation. While foundational, the practical application in RNA design often requires conditional generation (e.g., based on sequence constraints, secondary structure motifs, or functional sites). The pathways to effectively integrate such conditions into the RNA-FrameFlow framework are mentioned but remain largely unexplored in this submission.

---

> ### Author Response · Authors · 2025-05-26
> **Response to Reviewer dYWX**
>
> Dear Reviewer dYWX,
>
> Thank you for your detailed comments. We hope to address your concerns and suggested changes. **We have also uploaded an updated manuscript PDF. Changes and additions are colored red.**
>
> > Expanded Analysis of Evaluation Tool Limitations: While the appendix touches on RhoFold's biases, integrate a more prominent discussion in the main text about how the limitations of gRNAde and RhoFold might affect the interpretation of the reported "validity" and self-consistency scores. This adds a layer of critical self-assessment.
>
> In Appendix A.2, we have done a rigorous self-assessment of our evaluation pipeline. We provide an upper bound on the performance boost gained from integrating gRNAde and RhoFold. We imagine training samples from RNAsolo to be AI-generated and pass them through our evaluation pipeline, giving us a validity score of 43.7%. If we then pass our RNA-FrameFlow-generated samples through the same evaluation pipeline, we get a validity of 41%. This indicates our model is able to design samples that approximate the training distribution to a fair extent, despite the errors accumulated from using gRNAde and RhoFold. gRNAde (published in ICLR’25) has a wet-lab-validated success rate of 50% (compared to physics-based Rosetta with 35%). There is no other appropriate, wet-lab validated RNA inverse folding model.
>
> As you’ve highlighted, RhoFold has biases, which we document in Appendix A.3. To that end, we have also tried using Chai-1, an open-source structure prediction model that reaches AlphaFold3-level performance on nucleic acids, in our self-consistency pipeline. We provide this ablation in Appendix B.3. Our validity score with Chai is 39%.
>
> Additionally, to clarify, self-consistency is our metric FOR validity, ie, the percentage of structures that surpass the self-consistency TM-score threshold of 0.45 is called validity. We define it this way since there is no notion of designability for AI-generated RNA yet, compared to proteins. The value of 0.45 is chosen because two RNA folds with a TM-score of >= 0.45 are considered structurally similar.
>
> > Acknowledge Backbone-Only Limitations More Directly in Main Results: When discussing the generated structures and their novelty, more explicitly connect the backbone-only nature of the model to its potential limitations in capturing folds heavily reliant on specific base-mediated interactions. This manages reader expectations regarding the types of structures the current model can realistically generate.
>
> In Section 5 on Limitations and Discussions, we now include this in further detail in the subsection titled “Physical Violations”. We describe how our method primarily operates on the RNA backbone. We bolster the original description of base pairing and base stacking interactions by mentioning how our framework’s design choice of not including base atoms (ie, AUCG) might result in an inability to model complex base-mediated folds. However, we believe RNA-FrameFlow modeling the N1/N9 atom still allows it to uncover certain types of tertiary folds like pseudo-knots and partial base stacking. We provide visual examples in Figure 6.
>
> > Briefly Outline Future Directions for Conditional Generation: While a full exploration is beyond scope, a short paragraph outlining potential strategies or initial thoughts on how conditional information (e.g., secondary structure constraints) could be incorporated into the RNA-FrameFlow framework would be beneficial for readers interested in practical design applications.
>
> In Section 2.2, under the heading of “Conditional Generation”, we have added details of possible extensions for conditional generation using our RNA-FrameFlow as a base. We talk about inference-time guidance strategies, finetuning, and potential use of language model embeddings (eg: from ESM, RibonanzaNet, or RiNALMo) as additional features. Applications include RNA Aptamer design, conditioned on target protein structures and binding sites, which is not the focus of this work, but offer avenues for future works to tackle.
>
> We hope we have adequately clarified any doubts and appreciate the feedback!

---

### Review · Reviewer_fANp · 2025-05-11

**Summary Of Contributions:**

RNA-FRAMEFLOW is **claimed to be the first generative framework** for building 3-D RNA backbones from scratch. By extending the SE(3) flow-matching approach—originally crafted for proteins—the authors tailor the method to cope with RNA’s higher flexibility and 13-atom backbone. The backbone is encoded as rigid-body frames, while extra auxiliary losses smooth the training. A variety of evaluation metrics checks both self-consistency and physical plausibility, and new data augmentation tricks boost the variety and novelty of samples.

**Audience:**

Yes

**Claims And Evidence:**

Yes

**Requested Changes:**

I would like to see a more thorough discussion of the novelty and additional technique innovations that are introduced as compared to

[1] Yim, Jason, et al. "Fast protein backbone generation with SE (3) flow matching."
[2] RoseTTAFoldNA

I am not demanding a ground breaking innovations to be distilled out of the comparison but rather a more clear and dedicated section or perhaps a table outlining similarity and RNA specific adaptation.

Other than the above demand, i think this is a solid piece of work and meet TMLR's criteria for acceptance.

**Strengths And Weaknesses:**

**Strength**

1. Introduces the very first generative model dedicated to 3-D RNA backbone design.

2. Extends SE(3) flow matching from proteins to RNA by adding RNA-specific frame parameters and torsion-based atom placement

3. Builds a full metric suite (global TM/EMD, local angle plots, clash checks, self-consistency, etc.), which may offer the community a solid benchmark for future RNA generators.

4. Demonstrates solid performance on common RNA families (tRNA, 5S rRNA, …) and provides thorough ablations plus some discussion of remaining challenges.

5. Clear writing and intuitive figures

**Weakness**
1. After only a preliminary search, I found this paper "A Hyperbolic Discrete Diffusion 3D RNA Inverse Folding Model for functional RNA design" published recently that seem also leverage inverse folding for 3D and functional RNA design. This might invalidate author's claim of first generative model for such task. Perhaps, it would be useful to add discussion of this paper as well and highlight the distinction and novelty as compared to this work
2. The methodological novelty here feels modest. The authors simply apply FrameFlow to RNA backbone design, but using FrameFlow to produce frames isn’t new—and RoseTTAFoldNA has already shown how to frame RNA structures. So this looks more like an incremental mash-up than a breakthrough worthy of a flagship conference. To stand out, the paper should surface deeper insights—e.g., unique hurdles when porting FrameFlow to RNA and concrete solutions. While the authors mention data scarcity and frame representation, those points aren’t fresh: limited 3-D RNA data is already well known, and the proposed fixes—structural clustering and cropping—are routine techniques, not strong innovations in themselves. However, the acceptance criteria of TMLR does not particularly emphasize on novelty. Therefore, this is not a major weakness that drives my decision making
3. Because the cropping augmentation chops sequences in ways nature never would, it may sometimes spits out fragments that can’t fold

---

> ### Author Response · Authors · 2025-05-26
> **Response to Reviewer fANp**
>
> Dear Reviewer fANp,
>
> Thank you for your comments! We hope to address your concerns below. **We have also uploaded an updated manuscript PDF. Changes and additions are colored red.**
>
> > After only a preliminary search, I found this paper "A Hyperbolic Discrete Diffusion 3D RNA Inverse Folding Model for functional RNA design" published recently that seem also leverage inverse folding for 3D and functional RNA design. This might invalidate author's claim of first generative model for such task. Perhaps, it would be useful to add discussion of this paper as well and highlight the distinction and novelty as compared to this work.
>
> Indeed, there are several works that address the overall umbrella of “RNA design”. The preprint you mentioned is tackling RNA inverse design – given a 3D backbone, design RNA sequences for it. We are tackling RNA backbone design – generating physically realistic 3D backbones. We are the first work to tackle RNA backbone design.
>
> There has also been significant follow-up work on RNA backbone design based on the code, datasets, and artifacts we have released for RNA-FrameFlow. We provide a discussion in Section 6.
>
> > The methodological novelty here feels modest. The authors simply apply FrameFlow to RNA backbone design, but using FrameFlow to produce frames isn’t new—and RoseTTAFoldNA has already shown how to frame RNA structures. So this looks more like an incremental mash-up than a breakthrough worthy of a flagship conference. To stand out, the paper should surface deeper insights—e.g., unique hurdles when porting FrameFlow to RNA and concrete solutions. While the authors mention data scarcity and frame representation, those points aren’t fresh: limited 3-D RNA data is already well known, and the proposed fixes—structural clustering and cropping—are routine techniques, not strong innovations in themselves. However, the acceptance criteria of TMLR does not particularly emphasize on novelty. Therefore, this is not a major weakness that drives my decision making. Because the cropping augmentation chops sequences in ways nature never would, it may sometimes spits out fragments that can’t fold.
>
> Through this work, we are primarily interested in investigating whether existing protein design protocols can be adapted for realistic RNA design. Our answer is mostly positive, but we believe RNA data hasn’t hit critical mass yet to purely rely on scale. This requires us to seek inspiration from medicinal chemistry and structural biology literature to build RNA-specific inductive biases, which we hope to highlight as our novelty.
>
> 1. RoseTTAFold2NA is a method for structure prediction, which requires different inductive biases from structure generation/design/sampling. Our RNA frames are more principled: we choose {C4’, C3’, O4’} for the frame because these atoms are prone to minimal shifting and fluctuations in X-ray Crystallography scans (Please see our new Appendix C.5). This ensures the frame atoms are placed more confidently by our method. Additionally, to model the higher conformational diversity in RNA (than in proteins) to capture higher-order motion (eg: ring puckering; please see new Appendix C.6), we parameterize the other atoms via torsion angles instead of placing their atoms directly through 3D coordinates. This allows for efficient and realistic all-backbone-atom design of 3D RNA backbones compared to methods operating on raw coordinates.
>
> 2. We’ve adapted our auxiliary losses to help the model implicitly learn base pairing and base stacking interactions while ensuring the atoms within a nucleotide have minimal steric clashes and form realistic 3D poses along the generated chain.
>
> 3. Our custom self-consistency pipeline has been set up explicitly for RNA design. Relying on the medicinal chemistry literature, we’ve identified TM-score to be a better indicator of structural similarity which factors in the high structural flexibility of RNA compared to RMSD, which excessively penalizes two structures. As RNA has no notion of designability like proteins do, we instead refer to “validity” instead of “designability”. Our chosen TM-score threshold of 0.45 corresponds to two RNA folds being globally similar.
>
> While we agree our data augmentations are naive first efforts to addressing the paucity of 3D RNA structures, we still observe improved novelty and diversity in our sampled structures. Finding a middle ground between the three metrics (validity, novelty, diversity) is also a function of the self-consistency pipeline used. We provide an empirical upper bound on our self-consistency pipeline in Appendix A.2.
>
> We hope we have adequately clarified any doubts and appreciate the feedback!

---

### Review · Reviewer_Ym8X · 2025-05-21

**Summary Of Contributions:**

This work delves into RNA backbone structure generation, introducing the RNA-FrameFlow approach. The paper's key contributions include advancements in data preparation, evaluation protocols, and modeling techniques for RNA structures, focusing on the use of frames. From an algorithmic standpoint, the work is primarily centered around SE(3) flow matching, a technique extensively utilized in various domains such as protein design. The paper offers a wealth of content, providing a thorough analysis of the results and a clear discussion regarding its limitations.

**Audience:**

Yes

**Broader Impact Concerns:**

None.

**Claims And Evidence:**

Yes

**Requested Changes:**

Please see the weaknesses above.
The first two points would influence my recommendation. The third one would strengthen the work.

**Strengths And Weaknesses:**

Strengths:
- The paper provides details of data preparation and evaluation protocols, which I think is one of the main contributions of this paper.
- The authors have discussed how they define the RNA backbone frames and discuss the motivation behind this. Nice illustrations are also presented to help readers better understand the definition.
- The paper has designed several evaluation metrics, which I think are all rational.
- The paper has discussed limitations, which I think is good for community development and helps readers to better understand the methods.

Weakness:
- The baseline, MMDiff, is not designed for RNA-only structure, though it supports this in principle. So the comparison might be slightly unfair. I think a better baseline is atom-level flow models trained on same data.
- The authors explicitly discuss physical violations within the limitations, highlighting all-atom clashes. However, this raises the question of whether "all-atom" specifically refers to backbone atoms, as only frames and backbone torsions are predicted. What about the side-chains?
- (Note: this point does not influence my assessment.) When discussing the choice of frame atoms, the authors mentioned several motivations. Could the author provide more quantitative evidence for this? For example, "the chosen atoms spatially shift the least in naturally occurring RNA". This matters because the definition of frames determine the upperbound performance of this method.

---

> ### Author Response · Authors · 2025-05-26
> **Response to Reviewer Ym8X**
>
> Dear Reviewer Ym8X,
>
> Thank you for your comments! We hope to address your concerns below. **We have also uploaded an updated manuscript PDF. Changes and additions are colored red.**
>
> > The baseline, MMDiff, is not designed for RNA-only structure, though it supports this in principle. So the comparison might be slightly unfair. I think a better baseline is atom-level flow models trained on same data.
>
> Indeed, the original version of MMDiff is not meant for RNA-only structure generation and includes proteins and DNA. So, we retrained MMDiff on our training split from RNAsolo to offer a fair comparison to RNA-FrameFlow, which we’ve highlighted in Section 4.1. The results in Table 2 also document this retrained model, showing RNA-FrameFlow outperforming MMDiff in local structural realism.
>
> We agree that using all-atom flow models would act as fair baselines. However, since RNA nucleotides have 13 backbone atoms, we believe the large degrees of freedom (N_nt * 13 * 3) would stump most generative frameworks (flow matching, diffusion, rectified flows, etc), possibly yielding suboptimal results. We can neither rely on scale because high-resolution RNA 3D structure data is too scarce, nor on the most expressive neural network denoiser/vector field.
>
> > The authors explicitly discuss physical violations within the limitations, highlighting all-atom clashes. However, this raises the question of whether "all-atom" specifically refers to backbone atoms, as only frames and backbone torsions are predicted. What about the side-chains?
>
> We do not do sidechain packing at the moment and have revised the manuscript to clarify that we refer to all backbone atoms instead of ‘all-atom’ modelling. RNA-FrameFlow currently places the {C4’, C3’, O4’, P, OP1, OP2, C5’, C2’, C1’, O2’, O3’, N1/N9} atoms that constitute the RNA nucleotide backbone. This is analogous to protein backbone design, where the {CA, C, N, O} atoms are placed.
>
> To do RNA sidechain packing, we would have to know the identities of the nucleotide bases (ie, AUCG). Such a sequence-structure co-design task is beyond the scope of our work. However, protein-based methods like Multiflow [1] can likely be repurposed for this task.
>
> > When discussing the choice of frame atoms, the authors mentioned several motivations. Could the author provide more quantitative evidence for this? For example, "the chosen atoms spatially shift the least in naturally occurring RNA". This matters because the definition of frames determine the upperbound performance of this method.
>
> This is a great suggestion! In our updated manuscript PDF, we’ve added a new section, Appendix C.5, titled “Atomic Displacement of Frame Atoms”. For each nucleotide backbone atom in {P, OP1, OP2, C4’, C3’, O4’, O5’, C5’, C1’, C2’, O2’, O3’}, we examine its B-factor, which is a measure of atomic displacement, ie, how much an atom “wiggles” in place when the structure is determined by X-ray crystallography or Cryo-EM. We include the table here for your reference:
>
> | Atom | Mean B-factor $\downarrow$ | Std. Dev. B-factor $\downarrow$ | Median B-factor $\downarrow$ |
> |:----:|:-------------:|:------------------:|:---------------:|
> |  C4' |     111.44    |       105.94       |      84.84      |
> |  C3' |     111.74    |       105.96       |      85.41      |
> |  O4' |     111.22    |       105.91       |      84.86      |
> |   P  |     114.03    |       105.50       |      88.69      |
> |  OP1 |     113.46    |       105.35       |      87.56      |
> |  OP2 |     112.35    |       105.57       |      86.11      |
>
> Compared to the phosphate-group-centric frame choice of {P, OP1, OP2} employed by RF2NA, our sugar-ring-centric frame design {C4’, C3’, and O4’} is prone to lesser collective atomic deviation, making it slightly easier for our denoiser to reason about coordinate placements. Furthermore, our choice of frame allows RNA-FrameFlow to implicitly learn the mechanics of ring puckering, which is important for mediating intermolecular interactions and complex formation. We provide further evidence on capturing puckering behaviour in our new Appendix C.6.
>
> We additionally refer the reviewer to [2] and [3], that show the ribose atoms to be more spatially stable than phosphate group atoms to represent coarse-grained RNA structure, and are necessary to capture nuanced motions like puckering.
>
> We hope we have adequately clarified any doubts and appreciate the feedback!
>
> [1] Generative Flows on Discrete State-Spaces: Enabling Multimodal Flows with Applications to Protein Co-Design. Campbell et al. (2024)
>
> [2] A new way to see RNA. Keating et al. (2011)
>
> [3] Coarse-grained modeling of RNA 3D structure. Dawson et al. (2016)

---

### Comment · Action_Editor_2DAB · 2025-06-03
**Check author response and revision**

Dear Reviewers,

Please kindly take time to check author response and update your review comments.

Regards,
AE

---

### Decision · Action_Editor_2DAB · 2025-07-13

**Recommendation:** Accept with minor revision

**Additional Comments:**

(1) Comments from Reviewer dYWX: "Specifically, a concise summary of the evaluation tool limitations (currently in the appendix) should be integrated into the main text for better transparency. More critically, my request to update the "Related Work" section to include and contextualize recent concurrent works was completely overlooked. This revision is essential to properly situate the paper within this rapidly evolving field."

(2) Since you plan to release the code, please make it available in a public GitHub repository.

**Audience:**

Yes

**Audience Explanation:**

The researchers who work on RNA structure prediction, AI for Science will be interested in this paper.

**Claims And Evidence:**

Yes

**Claims Explanation:**

The authors focus on unconditional RNA backbone generation. They propose a generative model for creating 3D RNA backbone structures without relying on traditional motif assembly or heuristic methods. The model uses a flow matching approach, representing each RNA nucleotide as a rigid frame, with the remaining backbone atoms positioned using predicted torsion angles.

For evaluation, the authors assess how closely the generated backbones resemble real RNA structures in terms of global fold consistency, local geometric features, diversity, and novelty compared to known structures. The proposed method is compared to a retrained version of MMDiff, a diffusion model baseline, and demonstrates better performance in generating valid and realistic RNA backbones.

While reviewers have pointed out that the technical novelty of this paper is limited, they also recognize that the work is well executed and have recommended it for acceptance. We believe that readers can benefit from the methods and findings presented in this paper.

---

> ### Author Response · Authors · 2025-07-16
> **Response to AE**
>
> Dear AE and Reviewers,
>
> We thank you for all the useful feedback and commentary on our work, RNA-FrameFlow. We've incorporated the final revisions and have updated the camera-ready PDF here. Our source code on GitHub has also been shared therein. Please let us know if we can clarify anything else or include any further changes to the manuscript.
>
> Regards,
> Authors of RNA-FrameFlow

---

> > ### Comment · Action_Editor_2DAB · 2025-07-21
> >
> > Would you like to point how you address the following question:
> >
> > (1) Comments from Reviewer dYWX: "Specifically, a concise summary of the evaluation tool limitations (currently in the appendix) should be integrated into the main text for better transparency. More critically, my request to update the "Related Work" section to include and contextualize recent concurrent works was completely overlooked. This revision is essential to properly situate the paper within this rapidly evolving field."

---

> > > ### Author Response · Authors · 2025-07-21
> > > **Response to AE**
> > >
> > > Hello AE,
> > >
> > > In our Related Works section, we include two additional citations for similar methods, namely RiboGen [1] and RNAbpFlow [2]. RiboGen also uses our evaluation pipeline for generated structures.
> > >
> > > >  Since the original release of RNA-FrameFlow and its source code in 2024,
> > > several methods employ a similar SE(3) flow matching setup. Rubin et al. (2025) incorporate an additional
> > > sequence track for RNA structure-sequence co-design, using the MultiFlow framework from Campbell et al.
> > > (2024). Tarafder & Bhattacharya (2025) condition the 3D structure generation on pre-determined sequences
> > > and secondary structural contact maps, enabling all-atom resolution with base identities.
> > >
> > > [1] Rubin et al. (2025). RiboGen: RNA Sequence and Structure Co-Generation with Equivariant MultiFlow
> > >
> > > [2] Tarafder & Bhattacharya (2025). RNAbpFlow: Base pair-augmented SE(3)-flow matching for conditional RNA 3D structure generation